# Deep FlexQP: Accelerated Nonlinear Programming via Deep Unfolding

**Alex Oshin**[*‡], **Rahul Vodeb Ghosh**[*], **Augustinos D. Saravanos**[†§], **Evangelos A. Theodorou**[*]
[*]Georgia Institute of Technology      [†]Massachusetts Institute of Technology

## Abstract

We propose **FlexQP**, an always-feasible convex quadratic programming (QP) solver based on an $\ell_1$ elastic relaxation of the QP constraints. If the original constraints are feasible, FlexQP provably recovers the optimal solution. If the constraints are infeasible, FlexQP identifies a solution that minimizes the constraint violation while keeping the number of violated constraints sparse. Such infeasibilities arise naturally in sequential quadratic programming (SQP) subproblems due to the linearization of the constraints. We prove the convergence of FlexQP under mild coercivity assumptions, making it robust to both feasible and infeasible QPs. We then apply deep unfolding to learn LSTM-based, dimension-agnostic feedback policies for the algorithm parameters, yielding an accelerated **Deep FlexQP**. To preserve the exactness guarantees of the relaxation, we propose a normalized training loss that incorporates the Lagrange multipliers. We additionally design a log-scaled loss for PAC-Bayes generalization bounds that yields substantially tighter performance certificates, which we use to construct an accelerated SQP solver with guaranteed QP subproblem performance. Deep FlexQP outperforms state-of-the-art learned QP solvers on a suite of benchmarks including portfolio optimization, classification, and regression problems, and scales to dense QPs with over 10k variables and constraints via fine-tuning. When deployed within SQP, our approach solves nonlinear trajectory optimization problems 4-16x faster than SQP with OSQP while substantially improving success rates. On predictive safety filter problems, Deep FlexQP reduces safety violations by over 70% and increases task completion by 43% compared to existing methods.

## 1 Introduction

Nonlinear programming (NLP) is a key technique for both large-scale decision making, where difficulty arises due to the sheer number of variables and constraints, and real-time embedded systems, which need to solve many NLPs with similar structure quickly and robustly. Within NLP, quadratic programming (QP) plays a fundamental role as many real-world problems in optimal control (Anderson & Moore, 2007), portfolio optimization (Markowitz, 1952; Boyd et al., 2013; 2017), and machine learning (Huber, 1964; Cortes & Vapnik, 1995; Tibshirani, 1996; Candes et al., 2008) can be represented as QPs. Furthermore, sequential quadratic programming (SQP) methods utilize QP as a submodule to solve much more complicated problems where the objective and constraints may be nonlinear and nonconvex, such as in nonlinear model predictive control (MPC) (Diehl et al., 2009; Rawlings et al., 2020), state estimation (Aravkin et al., 2017), and power grid optimization (Montoya et al., 2019). SQP itself can even be used as a subproblem for solving mixed integer NLPs (Leyffer, 2001) and large-scale partial differential equations (Fang et al., 2023).

However, a common difficulty with SQP methods occurs when the linearization of the constraints results in an infeasible QP subproblem, and a large amount of research has focused on how to repair or avoid these infeasibilities, e.g., (Fletcher, 1985; Izmailov & Solodov, 2012), among others. A significant advantage of SNOPT (Gill et al., 2005), one of the most well-known SQP-based methods, is in its infeasibility detection and reduction handling. These considerations necessitate a fast yet robust QP solver that works under minimal assumptions on the problem parameters.

---

[‡]Contact: `alexoshin@gatech.edu`.
[§]Work done while at Georgia Tech.

To this end, we propose **FlexQP**, a *flexible* QP solver that is always-feasible, meaning that it can solve any QP regardless of the feasibility of the constraints. Our method is based on an exact relaxation of the QP constraints: if the original QP is feasible, then FlexQP identifies the optimal solution. On the other hand, if the original QP is infeasible, instead of erroring or failing to return a solution, FlexQP automatically identifies the infeasibilities while simultaneously finding a point that minimizes the constraint violation. This allows FlexQP to be a robust QP solver in and of itself, but its power shines when used as a submodule in an SQP-type method, see Fig. 1.

Moreover, through the relaxation of the constraints, multiple hyperparameters are introduced that can be difficult to tune and have a non-intuitive effect on the optimization. To address this shortcoming, we use deep unfolding (Monga et al., 2021) to design lightweight feedback policies for the parameters based on actual problem data and solutions for QP problems of interest, leading to an accelerated version titled **Deep FlexQP**. Learning the parameters in a data-driven fashion avoids the laborious process of tuning them by hand or designing heuristics for how they should be updated from one iteration to the next. Meanwhile, these data-driven rules have been shown to strongly outperform the hand-crafted ones, such as in the works by Ichnowski et al. (2021) and Saravanos et al. (2025).

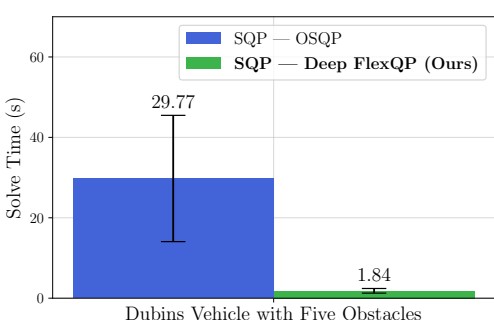

Figure 1: SQP with Deep FlexQP can solve highly constrained nonlinear optimizations over 16x faster than SQP with OSQP (averaged over 100 problems).

We thoroughly benchmark Deep FlexQP against traditional and learned QP optimizers on multiple QP problem classes including machine learning, portfolio optimization, and optimal control problems. Moreover, we certify the performance of Deep FlexQP through probably approximately correct (PAC) Bayes generalization bounds, which provide a guarantee on the mean performance of the optimizer. We propose a log-scaled training loss that better captures the performance of the optimizer when the residuals are very small. Finally, we deploy Deep FlexQP to solve nonlinearly constrained trajectory optimization and predictive safety filter problems (Wabersich & Zeilinger, 2021). Overall, Deep FlexQP can produce an order-of-magnitude speedup over OSQP (Stellato et al., 2020) when deployed as a subroutine in an SQP-based approach (Fig. 1), while also robustly handling infeasibilities that may occur due to a poor linearization or an over-constrained problem.

## 2    MOTIVATION & RELATED WORK

SQP solves smooth nonlinear optimization problems of the form

$$\underset{x}{\text{minimize}} \ \ f(x), \quad \text{subject to} \ \ g(x) \leq 0, \ \ h(x) = 0, \tag{1}$$

where $f : \mathbb{R}^n \to \mathbb{R}$ twice-differentiable is the objective to be minimized and $g : \mathbb{R}^n \to \mathbb{R}^m$ and $h : \mathbb{R}^n \to \mathbb{R}^p$ twice-differentiable describe the inequality and equality constraints, respectively. SQP solves Eq. (1) by solving a sequence of QP problems that are produced by linearizing the constraints and quadraticizing the Lagrangian[1] $\mathcal{L}(x, y_I, y_E) := f(x) + y_I^\top g(x) + y_E^\top h(x)$ around the current iterate $(x^k, y_I^k, y_E^k)$, where $y_I \in \mathbb{R}_+^m$ and $y_E \in \mathbb{R}^p$ are the dual variables for the inequality and equality constraints, respectively. This results in QP subproblems of the form:

$$\underset{dx}{\text{minimize}} \ \ \frac{1}{2} dx^\top \nabla_x^2 \mathcal{L}(x^k, y_I^k, y_E^k) \, dx + \nabla f(x^k)^\top dx, \tag{2a}$$

$$\text{subject to} \ \ g(x^k) + \partial g(x^k) dx \leq 0, \ \ h(x^k) + \partial h(x^k) dx = 0. \tag{2b}$$

Notably, the linearization of the constraints $g$ and $h$ may not produce a QP subproblem that is feasible, meaning that there may not exist any $dx$ that satisfies the linearized constraints Eq. (2b).

---

[1]In practice, the Hessian of the Lagrangian $\nabla_x^2 \mathcal{L}$ may be indefinite and is often replaced by a positive definite approximation, e.g., via a quasi-Newton update.

In this case, the SQP solver either terminates with a suboptimal point or a specialized routine needs to be run in order to reduce the infeasibility. For example, when SNOPT encounters an infeasible subproblem, it enters *elastic mode* and solves a new optimization where the constraints are relaxed using $\ell_1$ penalty functions (Gill et al., 2005). This is advantageous over other choices, such as an $\ell_2$ penalty, as the $\ell_1$ norm encourages sparsity in the constraint violation. This means the optimizer can naturally identify the constraints that are the most difficult to satisfy. In the context of mixed integer NLPs, infeasibilities are very likely to occur during the branch and bound process, so their fast and robust identification is crucial (Gill & Wong, 2011).

Moreover, as the interest in data-driven optimization grows, we often wish to solve many optimization problems with similar structure repeatedly (Amos et al., 2023), and potentially in parallel or in a batched fashion, such as in the method by Fang et al. (2023), which requires solving coupled systems of SQP problems in parallel. Needing to run specialized procedures in these cases is not scalable. Furthermore, identifying the best hyperparameters for each individual optimization problem is difficult and time consuming. Deep unfolding is a learning-to-optimize approach (Chen et al., 2022b) that has roots in the signal and image processing domains (Gregor & LeCun, 2010; Wang et al., 2015) and utilizes data-driven machine learning to reduce the cost of hyperparameter tuning and to accelerate convergence of model-based optimizers (Monga et al., 2021). It constitutes the state-of-the-art approach for sparse recovery (Liu et al., 2019) and video reconstruction (De Weerdt et al., 2024). In the context of NLP, deep unfolding has been recently applied to accelerate QPs. Saravanos et al. (2025) use an analogy to closed-loop control and learn feedback policies for the parameters of a deep-unfolded variant of the operator splitting QP (OSQP) solver (Stellato et al., 2020), which is a first-order method based on the alternating direction method of multipliers (ADMM) (Boyd et al., 2011). Their method can achieve orders-of-magnitude improvement in wall-clock time compared to OSQP, and they also propose a decentralized version for quickly solving QPs with distributed structure. Their idea is similar in vein to that of Ichnowski et al. (2021), who use reinforcement learning to train a policy that outputs the optimal parameters for OSQP, with the goal of accelerating the optimizer. Another related approach learns to warm-start a Douglas-Rachford splitting QP solver, with the goal of improving convergence speed (Sambharya et al., 2023).

Based on these considerations, in Section 3 we propose FlexQP, an always-feasible QP optimizer, and in Section 4 we apply deep unfolding, leading to Deep FlexQP. A more comprehensive survey of related work is provided in Appendix A.

## 3 FLEXQP: AN ALWAYS-FEASIBLE QUADRATIC PROGRAMMING SOLVER

Our proposed QP solver, FlexQP, transforms the QP constraints using an exact relaxation and then solves the resultant problem using an operator splitting inspired by OSQP (Stellato et al., 2020). We will assume the reader is familiar with ADMM; a good overview is provided by Boyd et al. (2011).

### 3.1 QUADRATIC PROGRAMMING

We are interested in solving convex QPs of the general form

$$\underset{x}{\text{minimize}} \quad \frac{1}{2}x^\top P x + q^\top x, \quad \text{subject to} \quad Gx \le h, \ Ax = b, \tag{3}$$

where $x \in \mathbb{R}^n$ is the decision variable. The objective is defined by the symmetric positive semidefinite quadratic cost matrix $P \in \mathbb{S}^n_+$ and the linear cost vector $q \in \mathbb{R}^n$. The inequality constraints are defined by the matrix $G \in \mathbb{R}^{m \times n}$ and the vector $h \in \mathbb{R}^m$. Similarly, the equality constraints are defined by the matrix $A \in \mathbb{R}^{p \times n}$ and vector $b \in \mathbb{R}^p$.

The optimality conditions for Eq. (3) are given by:

$$Px + q + G^\top y_I + A^\top y_E = 0, \ Gx - h \le 0, \ Ax - b = 0, \ y_I \odot (Gx - h) = 0, \ y_I \ge 0, \tag{4}$$

where $y_I \in \mathbb{R}^m$ and $y_E \in \mathbb{R}^p$ are the dual variables for the inequality and equality constraints, respectively. A tuple $(x^*, y_I^*, y_E^*)$ satisfying Eq. (4) is a solution to Eq. (3).

Throughout this work, we will avoid making any assumptions on $G$ or $A$, meaning that the constraints may be redundant, and in the worst case, there may not exist a feasible $x$ for the optimization. This allows for a unified way to handle infeasibilities when optimizations of the form Eq. (3) are embedded as a subproblem in SQP, as in Eq. (2).

### 3.2 ELASTIC FORMULATION

We start by introducing slack variables $s \in \mathbb{R}^m$, so Eq. (3) can be expressed equivalently as

$$\underset{x, s \geq 0}{\text{minimize}} \quad \frac{1}{2} x^\top P x + q^\top x, \quad \text{subject to} \ \ Gx + s - h = 0, \ \ Ax - b = 0. \tag{5}$$

This standard technique is the basis of many interior point algorithms (Nocedal & Wright, 2006). We then relax the set of equality constraints in Eq. (5) using $\ell_1$ penalty functions, yielding

$$\underset{x, s \geq 0}{\text{minimize}} \quad \phi(x, s; \mu_I, \mu_E) := \frac{1}{2} x^\top P x + q^\top x + \mu_I \left\| Gx + s - h \right\|_1 + \mu_E \left\| Ax - b \right\|_1, \tag{6}$$

with elastic penalty parameters $\mu_I, \mu_E > 0$. This relaxation approach is known as *elastic programming* (Brown & Graves, 1975), and one of the most well-known SQP-based solvers, SNOPT, uses this technique in order to reduce the infeasibility of a QP subproblem (Gill et al., 2005). This relaxation is also a fundamental step in the sequential $\ell_1$ quadratic programming method of Fletcher (1985). Notably, if Eq. (3) has a feasible solution and the elastic penalty parameters are sufficiently large, then the solutions to Eq. (3) and Eq. (6) are identical — this is why the relaxation is exact. On the other hand, if the original QP Eq. (3) is infeasible, then solving Eq. (6) finds a point that minimizes the constraint violation (Nocedal & Wright, 2006). This is formalized through the following theorem, which also describes what we mean by a sufficiently large penalty parameter.

**Theorem 3.1.** *Let* $(x^*, y_I^*, y_E^*)$ *solve Eq. (3). Let* $\mu_I^* = \|y_I^*\|_\infty$ *and* $\mu_E^* = \|y_E^*\|_\infty$. *Then, for all* $\mu_I \geq \mu_I^*$ *and* $\mu_E \geq \mu_E^*$, *the minimizers of Eq. (3) and Eq. (6) coincide.*

This theorem is a generalization of the one by Han & Mangasarian (1979) that shows we can select a different penalty parameter for the inequality vs. equality constraints. The proof, provided in Appendix B, relies on two simple facts: the optimality conditions of Eq. (3) and the convexity of the objective. The proof also shows that it is possible to select *vectors* of penalty parameters $\mu_I$ and $\mu_E$, as long as each $\mu_{I,i}$ and $\mu_{E,j}$ obeys the constraints $\mu_{I,i} \geq |y_{I,i}|$ or $\mu_{E,j} \geq |y_{E,j}|$, respectively.

**What happens when the penalty parameters $\mu$ do not satisfy the conditions of Theorem 3.1?** Using the interpretation of the Lagrange multiplier $y_i$ representing the cost of an individual constraint $i$ with associated penalty parameter $\mu_i > 0$, if $\mu_i \geq |y_i|$ then the penalty on violating constraint $i$ in Eq. (6) is large enough such that Theorem 3.1 holds. On the other hand, if $\mu_i < |y_i|$, then this constraint $i$ is not being penalized strong enough, and so the solution to Eq. (6) will violate this constraint, with the amount of violation proportional to the difference between $\mu_i$ and $|y_i|$. We use this interpretation in Section 4 to design feedback policies that select the best penalty parameters as a function of the optimizer state and enforce the condition $\mu_i \geq |y_i|$ during learning using a supervised loss that includes the Lagrange multipliers (see also Theorem 3.3 below).

### 3.3 OPERATOR SPLITTING & ADMM

The optimization in Eq. (6) can be simplified further to make the terms appearing in the $\ell_1$ penalty functions easier to handle. Introducing decision variables $z_I \in \mathbb{R}^m$ and $z_E \in \mathbb{R}^p$, we have

$$\underset{x, s, z_I, z_E}{\text{minimize}} \quad \frac{1}{2} x^\top P x + q^\top x + \mu_I \left\| z_I \right\|_1 + \mu_E \left\| z_E \right\|_1, \tag{7a}$$

$$\text{subject to} \quad z_I = Gx + s - h, \ \ z_E = Ax - b, \ \ s \geq 0. \tag{7b}$$

The variables $z_I$ and $z_E$ are equal to the constraint violation, so their optimal values can be viewed as a certificate of feasibility for Eq. (3) if $z_I^* = z_E^* = 0$ and infeasibility if $z_I^* \neq 0$ or $z_E^* \neq 0$. While it may seem tempting to apply ADMM to this formulation, the resultant updates will not have a closed-form solution no matter how the variable splitting is performed. Inspired by OSQP (Stellato et al., 2020), we perform a final transformation by introducing variables $\tilde{x}, \tilde{s}, \tilde{z}_I,$ and $\tilde{z}_E$, yielding

$$\underset{\tilde{\boldsymbol{x}}, \boldsymbol{x}}{\text{minimize}} \quad \underbrace{\frac{1}{2} \tilde{x}^\top P \tilde{x} + q^\top \tilde{x} + \mathcal{I}_I(\tilde{\boldsymbol{x}}) + \mathcal{I}_E(\tilde{\boldsymbol{x}})}_{=: f(\tilde{\boldsymbol{x}})} + \underbrace{\mathcal{I}_s(\boldsymbol{x}) + \mu_I \left\| z_I \right\|_1 + \mu_E \left\| z_E \right\|_1}_{=: g(\boldsymbol{x})}, \tag{8a}$$

$$\text{subject to} \quad \tilde{\boldsymbol{x}} = \boldsymbol{x}, \tag{8b}$$

where $\tilde{\boldsymbol{x}} = (\tilde{x}, \tilde{s}, \tilde{z}_I, \tilde{z}_E)$ and $\boldsymbol{x} = (x, s, z_I, z_E)$ for notational simplicity. The indicator functions $\mathcal{I}_I, \mathcal{I}_E$, and $\mathcal{I}_s$ are defined as

$$\mathcal{I}_I(\boldsymbol{x}) = \begin{cases} 0 & z_I = Gx + s - h, \\ +\infty & \text{otherwise,} \end{cases} \quad \mathcal{I}_E(\boldsymbol{x}) = \begin{cases} 0 & z_E = Ax - b, \\ +\infty & \text{otherwise,} \end{cases} \quad \mathcal{I}_s(\boldsymbol{x}) = \begin{cases} 0 & s \geq 0, \\ +\infty & \text{otherwise.} \end{cases}$$

Let the dual variables for the constraint $\tilde{\boldsymbol{x}} = \boldsymbol{x}$ be $\boldsymbol{y} = (w_x, w_s, y_I, y_E)$. The ADMM updates for solving Eq. (8) are given by

$$
\begin{aligned}
\tilde{\boldsymbol{x}}^{k+1} = \arg\min_{\tilde{\boldsymbol{x}}} \quad & f(\tilde{\boldsymbol{x}}) + (\sigma_x/2)\|\tilde{x} - x^k + \sigma_x^{-1}w_x^k\|_2^2 + (\sigma_s/2)\|\tilde{s} - s^k + \sigma_s^{-1}w_s^k\|_2^2 \\
& + (\rho_I/2)\|\tilde{z}_I - z_I^k + \rho_I^{-1}y_I^k\|_2^2 + (\rho_E/2)\|\tilde{z}_E - z_E^k + \rho_E^{-1}y_E^k\|_2^2,
\end{aligned}
\tag{9}
$$

$$x^{k+1} = \alpha\tilde{x}^{k+1} + (1-\alpha)x^k + \sigma_x^{-1}w_x^k, \qquad s^{k+1} = \left(\alpha\tilde{s}^{k+1} + (1-\alpha)s^k + \sigma_s^{-1}w_s^k\right)_+,$$

$$z_I^{k+1} = S_{\mu_I/\rho_I}(\alpha\tilde{z}_I^{k+1} + (1-\alpha)z_I^k + \rho_I^{-1}y_I^k), \quad z_E^{k+1} = S_{\mu_E/\rho_E}(\alpha\tilde{z}_E^{k+1} + (1-\alpha)z_E^k + \rho_E^{-1}y_E^k),$$

$$w_x^{k+1} = w_x^k + \sigma_x(\alpha\tilde{x}^{k+1} + (1-\alpha)x^k - x^{k+1}), \quad w_s^{k+1} = w_s^k + \sigma_s(\alpha\tilde{s}^{k+1} + (1-\alpha)s^k - s^{k+1}),$$

$$y_I^{k+1} = y_I^k + \rho_I(\alpha\tilde{z}_I^{k+1} + (1-\alpha)z_I^k - z_I^{k+1}), \quad y_E^{k+1} = y_E^k + \rho_E(\alpha\tilde{z}_E^{k+1} + (1-\alpha)z_E^k - z_E^{k+1}),$$

where $\sigma_x, \sigma_s, \rho_I, \rho_E > 0$ are the augmented Lagrangian penalty parameters, $\alpha \in (0, 2)$ is the ADMM relaxation parameter, $(s)_+ = \max(s, 0)$ is the rectified linear unit (ReLU) activation function, and $S_\kappa(z) = (z - \kappa)_+ - (-z - \kappa)_+$ is the soft thresholding operator, which is the proximal operator of the $\ell_1$ norm (Boyd et al., 2011). Since $x$ is unconstrained in the second block, we have that $w_x^{k+1} = 0$ for all $k \geq 0$, so the $w_x$ variable and update can be disregarded. The first block update is the most computationally-demanding step of the algorithm and requires the solution of an equality-constrained QP. We show how to solve this QP using either a direct or indirect method in Appendix C; the indirect method becomes the only suitable choice for large-scale problems where the dimension can be very large. The final algorithm is summarized in Algorithm 1 of Appendix D.

## 3.4 CONVERGENCE & THEORETICAL ANALYSIS

We establish the convergence of Algorithm 1 by showing that there always exists a saddle point of the Lagrangian for Eq. (8) for any $\mu_I, \mu_E > 0$, as long as the relaxed QP objective in Eq. (6) is not unbounded below. Then, since the optimization is a composite minimization of two closed, proper, convex functions, the algorithm converges by the general convergence of two-block ADMM (Boyd et al., 2011). The proof is provided in Appendix E.

**Theorem 3.2.** *Assume the relaxed objective $\phi(x, s; \mu_I, \mu_E)$, defined in Eq. (6), is not unbounded below. Then, Algorithm 1 for solving Eq. (8), equivalently Eq. (6), converges to a saddle point $(\hat{\tilde{\boldsymbol{x}}}, \hat{\boldsymbol{x}}, \hat{\boldsymbol{y}})$ of the Lagrangian for Eq. (8), given by*

$$\mathcal{L}(\tilde{\boldsymbol{x}}, \boldsymbol{x}, \boldsymbol{y}) = f(\tilde{\boldsymbol{x}}) + g(\boldsymbol{x}) + \boldsymbol{y}^\top(\tilde{\boldsymbol{x}} - \boldsymbol{x}). \tag{10}$$

The assumption that the objective is not unbounded below is called *coercivity* and is relatively weak (Bauschke & Combettes, 2017). Coercivity is only broken in the rare case when there exists an $x \neq 0$ such that $Px = 0$, $Gx = 0$, $Ax = 0$, and $q^\top x < 0$, which causes the optimal solution to diverge. In our cases of interest, we consider over-constrained problems with many more constraints than optimization variables, thus it is extremely unlikely that this assumption does not hold. Moreover, a bounded objective can be guaranteed if $P \succ 0$ or if $G$ or $A$ are full column rank.

The following theorem establishes the relationship between the FlexQP solution and the solution to the original QP and can be proven using the definition of soft thresholding. The proof is given in Appendix F.

**Theorem 3.3.** *For any $\mu_I, \mu_E > 0$, let $(\hat{\tilde{\boldsymbol{x}}}, \hat{\boldsymbol{x}}, \hat{\boldsymbol{y}})$ solve the relaxed QP Eq. (6) using Algorithm 1. Then, $|\hat{y}_{I,i}| \leq \mu_I$ for all inequality constraints $i = 1, \ldots, m$ and $|\hat{y}_{E,j}| \leq \mu_E$ for all equality constraints $j = 1, \ldots, p$. Furthermore, let $(x^*, y_I^*, y_E^*)$ solve Eq. (3), if it is feasible. If the conditions of Theorem 3.1 hold, then $(\hat{x}, \hat{y}_I, \hat{y}_E) = (x^*, y_I^*, y_E^*)$. Otherwise, for any infeasible constraint $i$ with associated dual variable $y_i$, the FlexQP solution satisfies $|\hat{y}_i| = \mu_i$.*

This shows that FlexQP solves Eq. (3) if the original QP is feasible, and otherwise identifies a stationary point of the infeasibilities, similar to Nocedal & Wright (2006, Theorem 17.4).

Finally, we summarize the key roles of the different hyperparameters of our algorithm. These insights are important for understanding the parameterization of our deep-unfolded architecture presented in the next section.

**Role of Elastic Penalty Parameters:** The parameters $\mu_I$ and $\mu_E$ only appear in a single step of the algorithm as part of the soft thresholding in the second block ADMM updates. Larger elastic penalties $\mu$ result in a larger threshold, meaning that a larger amount of constraint violation will be zeroed out. The choice of $\mu$ is key for satisfying the conditions of Theorem 3.1.

**Role of Augmented Lagrangian Penalty Parameters:** The role of the parameter $\sigma_x$ is to regularize the quadratic cost matrix $P$ and allow the equality-constrained QP Eq. (9) to admit a unique solution even if $P$ is not positive definite ($P = 0$ captures linear programs). We tested multiple fixed values of $\sigma_x$ along with adaptive and learned rules, but a fixed $\sigma_x = 1e{-}6$ appears to work very well in practice. This is similar to the choice of the $\sigma$ parameter in OSQP (Stellato et al., 2020).

The parameter $\sigma_s$ plays the role of a quadratic cost on the slack variable $s$ when solving Eq. (9). It also plays a small role in regularizing the constraint matrix $G$. In practice, tuning this parameter is the most difficult as the optimal value appears to depend strongly on the scaling of the objective and the constraints. This motivates our adaptive data-driven approach described in Section 4.

The penalty parameters $\rho_I$ and $\rho_E$ play two key roles in the algorithm. First, they regularize the constraint matrices $G$ and $A$ so that Eq. (9) is solvable regardless of the rank of $G$ or $A$. Second, they weight the noise level in the soft thresholding operations, playing an inverse role to $\mu_I$ and $\mu_E$, where a larger $\rho$ results in a smaller threshold. Determining the optimal values of these parameters by hand is unintuitive as they can have varying effects on the optimization, further motivating the deep unfolding approach presented in the next section.

## 4 ACCELERATING QUADRATIC PROGRAMMING VIA DEEP UNFOLDING

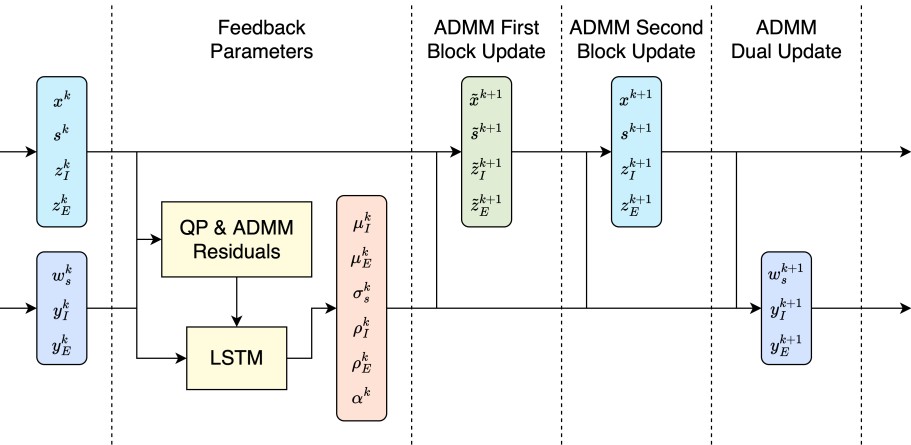

Figure 2: One layer of our proposed Deep FlexQP architecture. We learn dimension-agnostic feedback policies for the parameters while the propagation from one layer to the next is defined by the ADMM updates Eq. (9).

We focus our study on two recently proposed data-accelerated QP optimizers. The deep centralized QP optimizer from Saravanos et al. (2025) is a version of deep-unfolded OSQP where the penalty parameters $\rho$ and relaxation parameter $\alpha$ are learned as feedback policies on the problem residuals using an analogy to feedback control. In our comparisons, we refer to their method as **Deep OSQP**. The main limitation of their approach is that only scalar penalty parameters are learned, but it could be the case that different penalty parameters should be applied to different constraints to more effectively accelerate the optimizer. This was the main motivation for the deep QP method proposed by Ichnowski et al. (2021), where a policy that outputs a vector of penalty parameters is learned

using reinforcement learning. The vector policy is applied across the constraint dimensions so it is dimension-agnostic and generalizes across different problem classes. The authors show that the vector policy outperforms the scalar one across a suite of QP benchmarks. However, unlike Saravanos et al. (2025), the authors do not learn the relaxation parameter $\alpha$, which can greatly improve the convergence of ADMM (Boyd et al., 2011). We implement the approach from Ichnowski et al. (2021) and train it using the supervised learning scheme from Saravanos et al. (2025), leading to the baseline **Deep OSQP — RLQP Parameterization**. Finally, we implement a best-of-both-worlds approach that learns a vector feedback policy for the penalty parameters $\rho$ while also learning a policy for the ADMM relaxation parameter $\alpha$, which we call **Deep OSQP — Improved**.

## 4.1 DEEP FLEXQP ARCHITECTURE

Our proposed **Deep FlexQP** learns feedback policies for the algorithm parameters as a function of the current optimizer state as well as the QP and ADMM residuals (Fig. 2). We learn separate policies $\pi_I$, $\pi_E$, and $\pi_\alpha$ for the inequality constraints, equality constraints, and relaxation parameter, respectively. The constraint policies $\pi_I$ and $\pi_E$ take as input the ADMM variables along with the primal and dual QP residuals, and are applied in a batched fashion per constraint coefficient, making the architecture independent of problem size and permutation. The relaxation policy $\pi_\alpha$ takes as input the infinity norms of each residual, providing a scale-invariant representation of the convergence progress. Full expressions for the policy inputs and outputs are given in Appendix G.

All policies are parameterized by long short-term memory (LSTM) networks (Hochreiter & Schmidhuber, 1997), with the hypothesis that learning long-term dependencies can aid the selection of the optimal parameters. This furthers the idea from Saravanos et al. (2025) that time-varying feedback on the current (nominal) parameters can provide a large improvement. In our case, we are applying feedback based on a latent state capturing the optimization history. Our results show that LSTMs provide the most benefit for problems where the active constraints might change many times over the course of the optimization. An ablation analysis and further discussion is provided in Appendix N.

## 4.2 SUPERVISED LEARNING

For training Deep OSQP variants, we adopt the supervised learning approach from Saravanos et al. (2025). Let $x^k(\theta)$ be the $k^{\text{th}}$ iterate of Deep OSQP parameterized by $\theta$. The training objective is the weighted sum of the optimality gaps between the iterates $x^k(\theta)$ and the optimal solution $x^*$:

$$\underset{\theta}{\text{minimize}} \sum_{k=1}^{K} \gamma_k \left\| x^k(\theta) - x^* \right\|_2, \tag{11}$$

where $\gamma_k = \exp((k - K)/5)$ is a per-iteration scaling factor.

For training Deep FlexQP, we adopt a similar loss, but generalize it to incorporate the optimal Lagrange multipliers based on the discussion in Section 3. We also use the normalized optimality gaps instead of the unnormalized ones so that the scale is automatically determined based on the distance from the optimal solution:

$$\underset{\theta}{\text{minimize}} \sum_{k=1}^{K} \left\| \xi^k(\theta) - \xi^* \right\|_2 / \left\| \xi^* \right\|_2, \tag{12}$$

where $\xi = (x, y_I, y_E)$. By including the Lagrange multipliers here, we are able to enforce the Deep FlexQP optimizer to select penalty parameters that meet the conditions of Theorem 3.1, namely that $\mu_I \geq \|y_I^*\|_\infty$ and $\mu_E \geq \|y_E^*\|_\infty$. This is due to the fact that the Lagrange multipliers of Deep FlexQP $y_I(\theta)$ and $y_E(\theta)$ are upper-bounded (in absolute value) by the current selection of $\mu$ (see Theorem 3.3). An ablation studying the effect of this loss is provided in Appendix O.

## 4.3 PAC-BAYES GENERALIZATION BOUNDS

Recent approaches have been proposed for establishing generalization bounds for guaranteeing the performance of learning-to-optimize methods, including a binary loss approach from Sambharya & Stellato (2025) as well as a more informative progress metric by Saravanos et al. (2025), given as

$$L(\theta) = \min \left( \frac{\|x^K(\theta) - x^*\|_2}{\|x^0(\theta) - x^*\|_2}, 1 \right). \tag{13}$$

These approaches can be used to construct PAC-Bayes generalization bounds on the mean performance of the optimizer that hold with high probability. Nevertheless, a limitation of the resulting PAC-Bayes bound from Eq. (13) is that it assumes that the losses can fall anywhere within the range $[0, 1]$, despite the fact that, in practice, most of the final optimality gaps fall very close to 0 (on the order of $10^{-2}$ and smaller). In other words, the loss in Eq. (13) does not properly account for the scale of the errors, and as a result, obtaining a meaningful bound might require exponentially more training samples. For example, Fig. 4a shows that training for this generalization bound loss results in a bound that is uninformative since it sits above even the vanilla optimizers and does not capture the behavior well at small errors.

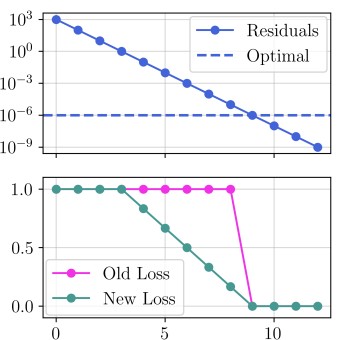

Figure 3: Log-scaled loss better captures small errors when the solution is close to the optimal.

To address this, we design a loss that is zero when the learner performs as well as or better than the optimal solution, and increases linearly as the performance decreases on a log-scale; see Fig. 3 for a visualization. Furthermore, we penalize distance from the optimal solution with respect to the norm of the QP residuals, which better takes into account the problem scale:

$$L(\theta) = \mathrm{clip}\left(1 - \frac{\log\|R(\xi^K(\theta))\|_2}{\log\|R(\xi^*)\|_2}, 0, 1\right), \tag{14}$$

where $R(\xi) := (Px + q + G^\top y_I + A^\top y_E, \max(Gx - h, 0), Ax - b)$ computes the residuals of the original QP in Eq. (3). As intended, training for this loss better captures the performance when the residuals are very small. Results are presented in Fig. 4b and Appendix M.

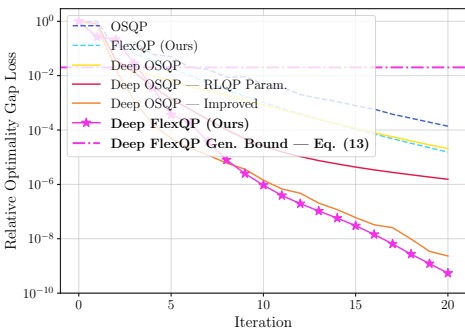
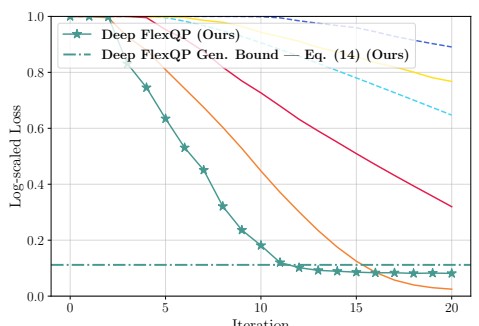

(a) Deep FlexQP trained for the generalization bound loss Eq. (13).

(b) Deep FlexQP trained for our proposed generalization bound loss Eq. (14).

Figure 4: Optimizer comparison on 1000 test LASSO problems. Training using our log-normalized loss Eq. (14) results in a substantially more informative performance guarantee.

## 5 EXPERIMENTAL RESULTS

We evaluate our approach on three classes of problems of increasing complexity: small- to medium-scale QPs, large-scale QPs, and nonconvex nonlinear programs solved via SQP. All algorithms were implemented using dense batched matrices in PyTorch, and all experiments were run on a system with an NVIDIA RTX 4090 GPU. Across all experiments, we use a learning rate of $10^{-3}$ and a batch size of 50 (except for the large-scale problems, which use a batch size of 1, as larger batches exceeded the available GPU memory).

### 5.1 SMALL- TO MEDIUM-SCALE QPS

We apply our deep-unfolding methodology to a benchmark suite of QPs including portfolio optimization problems from finance, classification and regression problems from machine learning, and

linear optimal control problems. The results are presented in Fig. 5, and details on the problem representations as well as the data generation processes are provided in Appendix H. In the following plots, **OSQP** and **FlexQP** are the best performing versions of OSQP and FlexQP, respectively, found using a hyperparameter search (details in Appendix J). **Deep OSQP** is the approach from Saravanos et al. (2025), **Deep OSQP — RLQP Parameterization** is the parameterization from Ichnowski et al. (2021), and **Deep OSQP — Improved** is the best-of-both-worlds version of deep-unfolded OSQP described in Section 4. Finally, **Deep FlexQP** is our proposed deep-unfolded FlexQP optimizer with LSTM policy parameterization and trained using the loss Eq. (12). Each model is trained for 500 epochs on 500 problems (except for the random QP classes, which use 2000 problems) and evaluated on 1000 test problems. We also perform an extensive timing comparison and analysis between all of the above optimizers, presented in Appendix J.

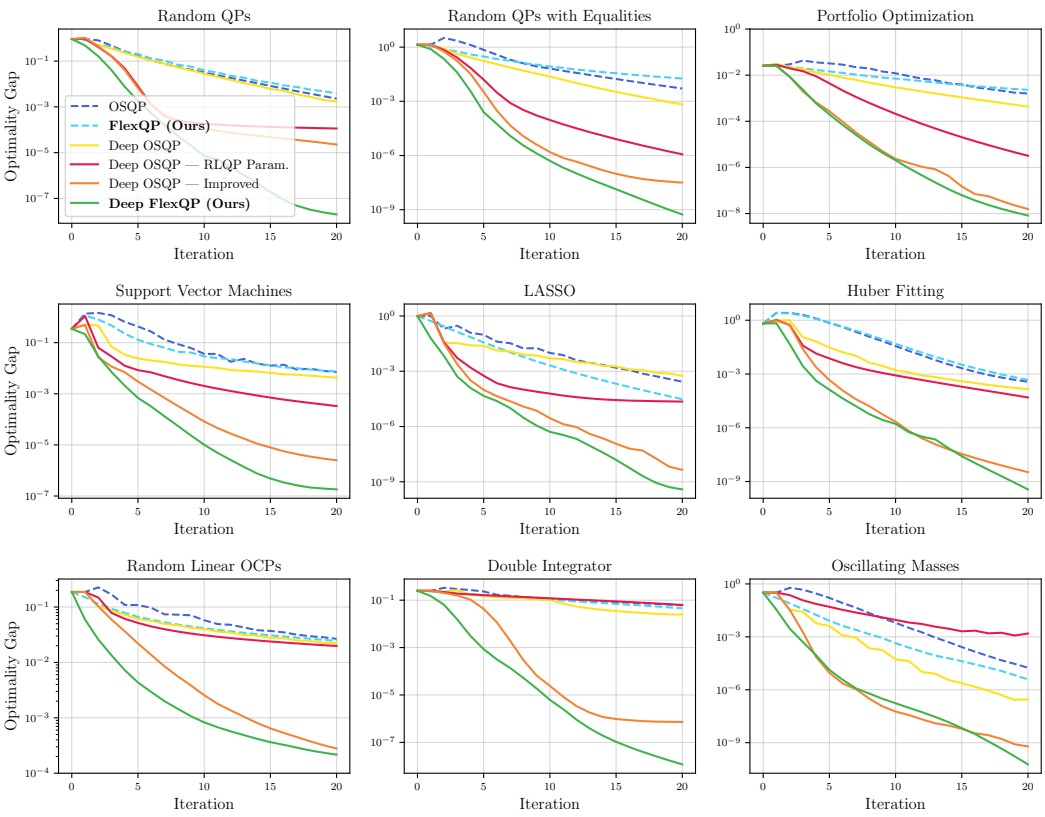

Figure 5: Performance comparison of learned deep optimizers and their non-learned counterparts. Our improved version of Deep OSQP outperforms the baselines, while Deep FlexQP consistently surpasses the rest of the methods in terms of convergence to the optimal QP solution.

## 5.2 LARGE-SCALE QPs

Next, we verify how well our methodology generalizes to large-scale QPs. In order to amortize the cost of training on these large-scale problems, we adopt a fine-tuning approach where the models trained on the small- to medium-scale problems from Section 5.1 are fine-tuned on a limited number of large-scale problems for a few epochs. Each model was fine-tuned on 100 training problems for 5 epochs. As each epoch takes roughly 3 hours to run, we estimate that training on the same number of problems and for the same number of epochs as the problems considered in Section 5.1 would take over 300 days, showing a clear benefit of the proposed fine-tuning approach.

Fig. 6 shows a comparison on 100 portfolio optimization and support vector machine (SVM) test problems with 10k variables and 10-20k constraints. Each optimizer is run until the infinity norm of the residuals falls below an absolute tolerance of $\varepsilon = 10^{-3}$, with a timeout of 10 minutes. Optimizers use the indirect method to solve their first block ADMM update (see Appendix C). For the portfolio

optimization problems, we report the average number of iterations each optimizer took to converge, and, for the SVM problems, we report the average number of conjugate gradient (CG) iterations that were necessary to solve the linear systems to a tolerance of $\varepsilon_{CG} = 10^{-2} \cdot \varepsilon$, with full results provided in Appendix K. We observe that the fine-tuned Deep FlexQP outperforms all of the other optimizers. Surprisingly, the fine-tuning approach does not seem to work as well for the Deep OSQP variants. Notably, we tried fine-tuning each of the methods using either the optimality gap loss Eq. (11) or the Lagrange multiplier loss Eq. (12). While the two losses yielded comparable performance on the portfolio optimization problems, the Lagrange multiplier loss performed significantly worse on the SVM problems for the Deep OSQP variants. As such, the reported Deep OSQP results use the models trained with the optimality gap loss Eq. (11). These results suggest that the superior fine-tuning performance of Deep FlexQP is attributable to the FlexQP architecture itself — specifically, the elastic relaxation and LSTM parameterization — rather than the choice of loss function.

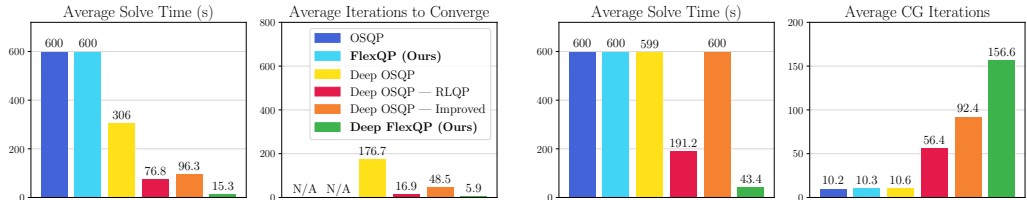

Figure 6: Learned vs. traditional optimizers on large-scale QPs. Left: portfolio optimization (10k variables, 10k constraints). Right: support vector machines (10k variables, 20k constraints).

## 5.3 NONCONVEX NONLINEAR PROGRAMMING USING SQP

Lastly, we apply Deep FlexQP as a submodule in SQP to solve nonconvex NLPs arising from nonlinear optimal control and nonlinear predictive safety filter problems. Training for the generalization bound loss Eq. (14) yields a numerical certificate of performance that we use when designing the SQP method. Results are shown in Fig. 1 and Fig. 7. Further details are provided in Appendix I.

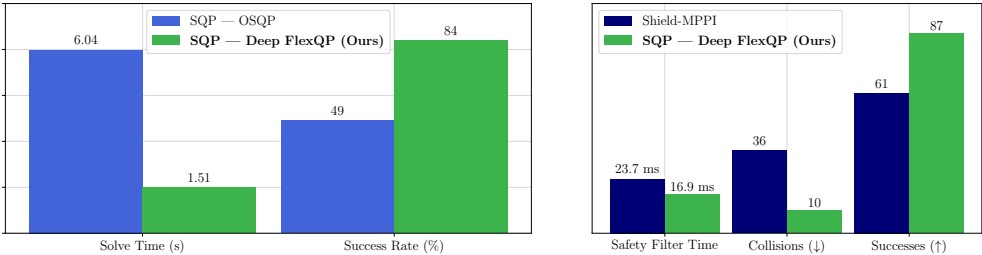

Figure 7: Comparison of our approach vs. traditional optimizer baselines on quadrotor trajectory optimization problems (left) and nonlinear predictive safety filter problems (right). Ours is faster than the baselines while vastly improving the task completion rate and safety.

## 6 CONCLUSION

We present FlexQP, a flexible QP solver that natively handles infeasible subproblems by minimizing constraint violation when no feasible solution exists. This property makes FlexQP particularly well-suited as a submodule in SQP, where infeasible QP subproblems are common and typically require ad hoc recovery strategies. Our accelerated variant, Deep FlexQP, outperforms both traditional solvers and learned approaches, converging in fewer iterations and less time per solve. Generalization bounds provide a numerical certificate of performance, and we use these bounds to design SQP solvers for nonlinear optimal control and predictive safety filters. In our SQP experiments, Deep FlexQP provides a substantial speedup over traditional approaches while also enabling graceful recovery from infeasibility without additional machinery. Promising future directions include learning warm-starts for FlexQP and extending it to the distributed QP setting of Saravanos et al. (2025).

ACKNOWLEDGMENTS

This work was supported by the Army Research Office Award #W911NF2010151. Alex Oshin was additionally supported by the Georgia Tech Aerospace Engineering Graduate Student Fellowship, and Augustinos Saravanos by the A. Onassis Foundation Scholarship.

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

## A  EXTENDED RELATED WORK

**Quadratic Programming.** Active-set methods, developed as extensions of the simplex method for QPs (Wolfe, 1959), can be efficiently warm-started but suffer from worst-case exponential complexity (Klee & Minty, 1970). Interior-point methods rose to prominence with the polynomial-time algorithms of Karmarkar (1984) and their extension to general convex programs via self-concordant barriers (Nesterov & Nemirovskii, 1994), replacing active-set methods as the dominant approach for small- to medium-scale problems (Wright, 2004). However, interior-point methods often do not scale well to high-dimensional problems and are difficult to warm-start. This has led to a growing interest in first-order methods in recent years. These methods are largely based on operator splitting methods such as ADMM, which is equivalent to Douglas-Rachford splitting applied to the dual (Gabay, 1983). First-order methods are appealing as they scale favorably to millions of variables and can be effectively warm-started, unlike interior-point methods. In addition, they are especially well-suited to applications such as MPC, where only moderate accuracy is required and the slow tail convergence of ADMM (He & Yuan, 2012) is less problematic.

SCS (O'Donoghue et al., 2016) is a first-order solver for general conic programs based on a homogeneous self-dual embedding. However, using this embedding is inefficient for QPs since it requires reformulating the quadratic cost as a second-order cone constraint. OSQP (Stellato et al., 2020) and COSMO (Garstka et al., 2021) are ADMM-based solvers specifically designed for QPs and general conic programs, respectively. COSMO can solve QPs as a special case, with the resulting algorithm being nearly identical to OSQP. We discuss these methods further below in the context of infeasibility detection.

The proposed FlexQP algorithm is a first-order method based on ADMM that has the same per-iteration complexity as OSQP and COSMO (see Appendix C), but uniquely handles infeasible QPs through an exact $\ell_1$ relaxation of the constraints (Section 3).

**Penalty Methods and Constrained Optimization.** Penalty methods have a long history in constrained nonlinear programming, starting from the early works by Courant (1943) proposing the quadratic penalty method, to the log and inverse barrier methods of Frisch (1955) and Carroll (1961). Fiacco & McCormick (1968) provided a unifying perspective on *exterior-point* penalty methods and *interior-point* barrier methods, and also began the analysis on primal-dual methods.

However, both exterior-point and interior-point methods suffer from poor numerical conditioning as the algorithm converges because the penalty parameter needs to asymptotically approach infinity (for exterior-point methods) or zero (for interior-point methods). This led to the development of exact penalty methods, based primarily on $\ell_1$ penalty functions (Eremin, 1966; Zangwill, 1967). These methods are more numerically stable because the penalty parameter need only be larger than the largest Lagrange multiplier. The necessary and sufficient conditions for exactness were given by Pietrzykowski (1969). The $\ell_1$ penalty function was also used by Han (1977) to prove the global convergence of SQP, foreshadowing the key role that exact penalties play in handling infeasible subproblems, as we discuss below. A modern discussion is given by Nocedal & Wright (2006) and we use these results to prove the exactness of our relaxed problem formulation (Theorem 3.1).

While exact penalty methods fixed the numerical ill-conditioning of previous approaches, the $\ell_1$ penalty function is non-differentiable at the boundary of the feasible region, resulting in a phenomenon known as the *Maratos effect* (Maratos, 1978). This sparked interest in methods that modify the SQP algorithm to preserve convergence, such as watchdog procedures (Chamberlain et al., 1982) and second-order corrections (Conn & Pietrzykowski, 1977; Fletcher, 1982).

**Operator Splitting and ADMM.** Concurrently, the augmented Lagrangian (AL) method, also known as the method of multipliers, was developed by Hestenes (1969) and Powell (1969). The alternating direction method of multipliers (ADMM) was subsequently developed by Glowinski & Marroco (1975) and Gabay & Mercier (1976). The popularity of ADMM increased significantly in the 2010s after its suitability for solving large-scale optimization problems was highlighted by Boyd et al. (2011). Since then, highly scalable ADMM-based methods have been developed for signal processing (Mateos et al., 2010; Zhang & Kwok, 2014), multi-agent robotics (Saravanos et al., 2021; 2023; Abdul et al., 2025), and power systems (Erseghe, 2014; Mhanna et al., 2018; Xu et al., 2018), among other domains.

Infeasibility identification for ADMM-based methods was a major concern, as it was established that the iterates of ADMM diverge if the problem is infeasible (Eckstein & Bertsekas, 1992). SCS is an ADMM-based method for general conic programs that solves a homogeneous self-dual embedding to identify infeasibilities (O'Donoghue et al., 2016). The work by Banjac et al. (2019) showed that the *differences* of the ADMM iterates are convergent and can be used to construct certificates of infeasibility for the original problem. This technique is used by both OSQP (Stellato et al., 2020) and COSMO (Garstka et al., 2021) to identify infeasible problems.

Unlike these approaches, FlexQP certifies infeasibility through its relaxation rather than as a separate detection mechanism, returning a minimum-violation solution directly. This addresses a weakness of the existing methods, as even though they can successfully identify infeasible problems, they do not return a useful approximate solution to the original problem that can be used by the user.

**Sequential Quadratic Programming.** Since its inception in 1963 (Wilson, 1963), SQP has evolved into a general methodology for solving complex and highly constrained NLP problems. A key challenge in SQP methods is that the QP subproblems formed by linearizing the constraints can become infeasible, even when the original NLP is feasible. This issue is particularly relevant for learning-based settings, where the iterative training procedure often causes intermediate QP subproblems to become infeasible.

Several strategies have been developed to handle such infeasibilities; a recent unified perspective of these methods is provided by Kiessling et al. (2025). One approach is to run a feasibility restoration phase to locate a feasible point when an infeasible QP is encountered (Nocedal & Wright, 2006), such as in the FilterSQP method by Fletcher & Leyffer (2002). However, these procedures require extensive tuning and are not scalable for batched settings. A complementary approach is to modify the QP subproblem itself to ensure that a meaningful step can always be computed. Burke & Han (1989) proposed a robust QP subproblem for this purpose, and SNOPT's elastic mode (Gill et al., 2005) famously solves a relaxed QP when the original subproblem is infeasible. Meanwhile, stabilized SQP methods (Wright, 1998; Hager, 1999; Izmailov & Solodov, 2012) construct modified QP subproblems using an augmented Lagrangian objective, providing a mechanism to handle the degeneracy of practical optimization.

At the 2011 Advances in Numerical Computation Workshop, Gill (2011) noted that "almost all practical optimization problems are degenerate," emphasizing the role that a robust QP solver plays in the field of NLP. Rather than handling infeasibilities and degeneracies on a per-problem basis, FlexQP unifies their treatment within a single batched solver, making it well-suited for learning-based SQP methods and GPU implementations. These considerations are particularly relevant for nonlinear MPC (Diehl et al., 2009; Rawlings et al., 2020) and safety-critical control (Ames et al., 2019; Wabersich & Zeilinger, 2021), where SQP is used in real-time control loops. A recent overview of constrained second-order trajectory optimization methods, including SQP and augmented Lagrangian-based approaches, is provided by Aoyama et al. (2024).

**Learning to Optimize and Deep Unfolding.** Learning to optimize or *amortized optimization* has established itself as a key technique for improving decision making using data-driven techniques (Chen et al., 2022b; Amos et al., 2023). Recent work has focused on understanding the representations (Liu et al., 2023), training dynamics (Metz et al., 2019; Scieur et al., 2022), and convergence (Sucker & Ochs, 2025) of learning-to-optimize approaches. These methods can be broadly classified into two main categories: model-free approaches that learn an optimizer from scratch, and model-based approaches that augment an existing algorithm with learnable components (Shlezinger & Eldar, 2023).

Deep unfolding, also known as *algorithm unrolling* (Monga et al., 2021), is a model-based learning-to-optimize approach that began with the seminal work by Gregor & LeCun (2010) on the learned iterative shrinkage and thresholding algorithm (LISTA). The LISTA framework has been extensively analyzed and improved over the years (Moreau & Bruna, 2017; Chen et al., 2018; Liu et al., 2019; Chen et al., 2021), and deep unfolding has since been applied to various fields, including speech processing (Hershey et al., 2014; Heigold et al., 2016), compressive imaging (Zhang & Ghanem, 2018; Mardani et al., 2018), and video reconstruction (Luong et al., 2021; De Weerdt et al., 2024), among others (Adler & Öktem, 2018; Gupta et al., 2018; Solomon et al., 2020). Deep unfolding has also been effectively applied to ADMM-based problems and algorithms (Yang et al., 2016; Ding et al., 2018; Xie et al., 2019; Yang et al., 2020; Saravanos et al., 2025).

Deep FlexQP is a deep-unfolded optimizer that learns hyperparameter policies from example problems of interest but maintains the mathematical structure from the underlying optimization process to inherit the guarantees and interpretability of the base optimizer.

**Learned Solvers for Quadratic Programming.** Learning-based approaches for quadratic programming have received considerable attention in recent years. Methods based on warm-starting determine a good initialization for a downstream solver using machine learning. Sambharya et al. (2023) learn a warm-start network for a Douglas-Rachford splitting QP solver, and the method was extended to general fixed-point algorithms in Sambharya et al. (2024). A related approach uses a deep-unfolded Douglas-Rachford splitting solver to warm-start SCS (Xiong et al., 2025).

A complementary class of approaches enhances the iterations of an optimization algorithm with learnable components. Deep FlexQP falls within this latter category of methods. Ichnowski et al. (2021) use reinforcement learning to learn dimension-agnostic policies for selecting the penalty parameters of OSQP. Saravanos et al. (2025) presented deep-unfolded optimization architectures for centralized and distributed QP, drawing an analogy to closed-loop control in order to design learned feedback policies for the hyperparameters of centralized and distributed OSQP. These approaches demonstrated remarkable scalability, solving problems with tens or even hundreds of thousands of variables to satisfactory accuracy under limited computational budgets. Similarly, Sambharya & Stellato (2024) learn step-varying and steady-state hyperparameters for several methods, including OSQP and SCS. We use the insight of dimension-independence from Ichnowski et al. (2021) along with the closed-loop control analogy from Saravanos et al. (2025) to learn component-wise LSTM policies for the hyperparameters of FlexQP. Consistent with the findings of Sambharya & Stellato (2024), the learned policies quickly adjust hyperparameters within the first few iterations before converging to a steady-state regime (see Appendix L).

**Performance Guarantees for Learned Optimizers.** Providing formal performance guarantees for learned optimizers remains an open challenge. Some asymptotic convergence guarantees have been derived using safeguarding mechanisms (Heaton et al., 2023) or by ensuring that the iterates do not deviate very far from a known convergent algorithm (Banert et al., 2024; Martin et al., 2025; Martin & Belgioioso, 2026). Other approaches ensure convergence through greedy objective descent (Fahy et al., 2024) or provide deterministic worst-case certificates via the performance estimation problem (PEP) framework (Sambharya et al., 2025).

Another class of guarantees relies on PAC-Bayes bounds (Sambharya & Stellato, 2025; Sucker et al., 2025; Saravanos et al., 2025; Sucker & Ochs, 2025), which provide certificates of performance that hold with high probability. However, the loss functions used in these works are based on binary objectives or linear ratios that saturate as the optimizer converges, yielding uninformative bounds when the learned optimizer achieves very low error. To address this, we develop our PAC-Bayes bounds based on a log-scaled, residual-based loss function that better captures the performance of the optimizer as the error gets arbitrarily small (see Section 4.3 and Appendix M).

## B   PROOF OF THEOREM 3.1

First, we state an equivalent representation of the relaxed QP:

**Lemma B.1.** *The relaxed QP Eq. (6) can equivalently be expressed as the following optimization:*

$$\min_x \quad \phi(x; \mu_I, \mu_E) := \frac{1}{2} x^\top P x + q^\top x + \mu_I \left\| (Gx - h)_+ \right\|_1 + \mu_E \left\| Ax - b \right\|_1. \tag{15}$$

*Proof.* Convert the optimization to the form without slack variables. This can be equivalently derived by directly relaxing the constraints of the original QP Eq. (3) using $\ell_1$ penalties. □

Now, the sketch of the proof of Theorem 3.1 is as follows. We will show for any $\mu_I \geq \mu_I^* = \|y_I^*\|_\infty$ and for any $\mu_E \geq \mu_E^* = \|y_E^*\|_\infty$ that

1. if $x^*$ solves Eq. (3), then $x^*$ solves Eq. (15), and

2. if $\hat{x}$ solves Eq. (15) then $\hat{x}$ solves Eq. (3).

**Part 1:** Let $x^*$ solve Eq. (3). For any $x \in \mathbb{R}^n$ we have that

$$\phi(x; \mu_I, \mu_E) = \frac{1}{2}x^\top P x + q^\top x + \mu_I \left\| (Gx - h)_+ \right\|_1 + \mu_E \left\| Ax - b \right\|_1 \tag{16a}$$

$$= \frac{1}{2}x^\top P x + q^\top x + \mu_I \sum_{i=1}^{m} (g_i^\top x - h_i)_+ + \mu_E \sum_{i=1}^{p} |a_i^\top x - b_i| \tag{16b}$$

$$\geq \frac{1}{2}x^\top P x + q^\top x + \|y_I^*\|_\infty \sum_{i=1}^{m} (g_i^\top x - h_i)_+ + \|y_E^*\|_\infty \sum_{i=1}^{p} |a_i^\top x - b_i| \tag{16c}$$

$$\geq \frac{1}{2}x^\top P x + q^\top x + \sum_{i=1}^{m} y_{I,i}^*(g_i^\top x - h_i)_+ + \sum_{i=1}^{p} y_{E,i}^*|a_i^\top x - b_i| \tag{16d}$$

$$\geq \frac{1}{2}x^\top P x + q^\top x + \sum_{i=1}^{m} y_{I,i}^*(g_i^\top x - h_i) + \sum_{i=1}^{p} y_{E,i}^*(a_i^\top x - b_i) \tag{16e}$$

$$= \frac{1}{2}x^\top P x + q^\top x + \sum_{i=1}^{m} y_{I,i}^*(g_i^\top x^* - h_i + g_i^\top(x - x^*)) \tag{16f}$$

$$+ \sum_{i=1}^{p} y_{E,i}^*(a_i^\top x^* - b_i + a_i^\top(x - x^*))$$

$$= \frac{1}{2}x^\top P x + q^\top x + \sum_{i=1}^{m} y_{I,i}^* g_i^\top(x - x^*) + \sum_{i=1}^{p} y_{E,i}^* a_i^\top(x - x^*) \tag{16g}$$

$$= \frac{1}{2}x^\top P x + q^\top x + (G^\top y_I^* + A^\top y_E^*)^\top(x - x^*) \tag{16h}$$

$$= \frac{1}{2}x^\top P x + q^\top x - (Px^* + q)^\top(x - x^*) \tag{16i}$$

$$= \frac{1}{2}x^{*\top} P x^* + q^\top x^* + \frac{1}{2}(x - x^*)^\top P(x - x^*) \tag{16j}$$

$$\geq \frac{1}{2}x^{*\top} P x^* + q^\top x^* \tag{16k}$$

$$= \frac{1}{2}x^{*\top} P x^* + q^\top x^* + \mu_I \left\| (Gx^* - h)_+ \right\|_1 + \mu_E \left\| Ax^* - b \right\|_1 \tag{16l}$$

$$= \phi(x^*; \mu_I, \mu_E). \tag{16m}$$

The step from Eq. (16b) to Eq. (16c) follows from the fact that $\mu_I \geq \|y_I^*\|_\infty$ and $\mu_E \geq \|y_E^*\|_\infty$, and Eq. (16d) follows from the definition of the infinity norm. Eq. (16e) follows from the definition of ReLU and the absolute value and Eq. (16f) is implied by the linearity of the constraints. Obtaining Eq. (16g) follows from the complementary slackness condition $y_{I,i}^*(g_i^\top x^* - h_i) = 0$ for all $i = 1, \ldots, m$, and feasibility $a_i^\top x^* - b_i = 0$ for all $i = 1, \ldots, p$. Eq. (16i) comes from the stationary condition $Px^* + q + G^\top y_I^* + A^\top y_E^* = 0$. Finally, obtaining Eq. (16k) follows from the positive semidefiniteness of $P$.

Thus, we have shown that $\phi(x^*; \mu_I, \mu_E) \leq \phi(x; \mu_I, \mu_E)$ for any $x$, which implies that $x^*$ minimizes $\phi(x; \mu_I, \mu_E)$ and therefore solves Eq. (15).

**Part 2:** Next, let $\hat{x}$ solve Eq. (15). If $x^* \neq \hat{x}$ solves Eq. (3), then we have that

$$\phi(\hat{x}; \mu_I, \mu_E) = \frac{1}{2}\hat{x}^\top P \hat{x} + q^\top \hat{x} + \mu_I \left\| (G\hat{x} - h)_+ \right\|_1 + \mu_E \left\| A\hat{x} - b \right\|_1 \tag{17a}$$

$$\leq \frac{1}{2}x^{*\top} P x^* + q^\top x^* + \mu_I \left\| (Gx^* - h)_+ \right\|_1 + \mu_E \left\| Ax^* - b \right\|_1 \tag{17b}$$

$$= \frac{1}{2}x^{*\top} P x^* + q^\top x^*, \tag{17c}$$

which follows by the optimality of $\hat{x}$. Now, assume that $\hat{x}$ is not feasible for Eq. (3). Then

$$\frac{1}{2}\hat{x}^\top P \hat{x} + q^\top \hat{x} \geq \frac{1}{2}x^{*\top} P x^* + q^\top x^* + (Px^* + q)^\top(\hat{x} - x^*) \tag{18a}$$

$$= \frac{1}{2}x^{*\top}Px^* + q^\top x^* - \sum_{i=1}^{m} y^*_{I,i}g_i^\top(\hat{x} - x^*) - \sum_{i=1}^{p} y^*_{E,i}a_i^\top(\hat{x} - x^*) \tag{18b}$$

$$= \frac{1}{2}x^{*\top}Px^* + q^\top x^* - \sum_{i=1}^{m} y^*_{I,i}(g_i^\top \hat{x} - h_i - (g_i^\top x^* - h_i)) \tag{18c}$$

$$- \sum_{i=1}^{p} y^*_{E,i}(a_i^\top \hat{x} - b_i - (a_i^\top x^* - b_i))$$

$$= \frac{1}{2}x^{*\top}Px^* + q^\top x^* - \sum_{i=1}^{m} y^*_{I,i}(g_i^\top \hat{x} - h_i) - \sum_{i=1}^{p} y^*_{E,i}(a_i^\top \hat{x} - b_i) \tag{18d}$$

$$\geq \frac{1}{2}x^{*\top}Px^* + q^\top x^* - \mu_I \sum_{i=1}^{m} g_i^\top \hat{x} - h_i - \mu_E \sum_{i=1}^{p} a_i^\top \hat{x} - b_i \tag{18e}$$

$$\geq \frac{1}{2}x^{*\top}Px^* + q^\top x^* - \mu_I \sum_{i=1}^{m}(g_i^\top \hat{x} - h_i)_+ - \mu_E \sum_{i=1}^{p}|a_i^\top \hat{x} - b_i|, \tag{18f}$$

where we have used the same facts as in Part 1. Rearranging, we have that

$$\frac{1}{2}\hat{x}^\top P\hat{x} + q^\top \hat{x} + \mu_I \left\|(G\hat{x} - h)_+\right\|_1 + \mu_E \left\|A\hat{x} - b\right\|_1 \geq \frac{1}{2}x^{*\top}Px^* + q^\top x^*, \tag{19}$$

but either $\hat{x} = x^*$ or this contradicts the fact that $\hat{x}$ minimized $\phi(\cdot; \mu_I, \mu_E)$ in Eq. (17c). Thus, $\hat{x}$ must be feasible for Eq. (3). Therefore, by Eq. (17c) we have that

$$\frac{1}{2}\hat{x}^\top P\hat{x} + q^\top \hat{x} \leq \frac{1}{2}x^{*\top}Px^* + q^\top x^*, \tag{20}$$

so $\hat{x}$ minimizes the quadratic objective and thus solves Eq. (3), completing the proof.

## C   FLEXQP — FIRST BLOCK ADMM UPDATE

The most computationally demanding step of FlexQP is the first block update Eq. (9), which is an equality-constrained QP:

$$\underset{\tilde{x},\tilde{s},\tilde{z}_I,\tilde{z}_E}{\text{minimize}} \quad \frac{1}{2}\tilde{x}^\top P\tilde{x} + q^\top \tilde{x} + (\sigma_x/2)\left\|\tilde{x} - x^k\right\|_2^2 + (\sigma_s/2)\left\|\tilde{s} - s^k + \sigma_s^{-1}w_s^k\right\|_2^2 \tag{21a}$$

$$+ (\rho_I/2)\left\|\tilde{z}_I - z_I^k + \rho_I^{-1}y_I^k\right\|_2^2 + (\rho_E/2)\left\|\tilde{z}_E - z_E^k + \rho_E^{-1}y_E^k\right\|_2^2,$$

$$\text{subject to} \quad \tilde{z}_I = G\tilde{x} + \tilde{s} - h, \quad \tilde{z}_E = A\tilde{x} - b. \tag{21b}$$

The optimality conditions for this QP are given by

$$P\tilde{x} + q + \sigma_x(\tilde{x} - x^k) + G^\top\tilde{\nu}_I + A^\top\tilde{\nu}_E = 0, \tag{22a}$$

$$\sigma_s(\tilde{s} - s^k) + w_s^k + \tilde{\nu}_I = 0, \tag{22b}$$

$$\rho_I(\tilde{z}_I - z_I^k) + y_I^k - \tilde{\nu}_I = 0, \tag{22c}$$

$$\rho_E(\tilde{z}_E - z_E^k) + y_E^k - \tilde{\nu}_E = 0, \tag{22d}$$

$$G\tilde{x} + \tilde{s} - \tilde{z}_I - h = 0, \tag{22e}$$

$$A\tilde{x} - \tilde{z}_E - b = 0, \tag{22f}$$

where $\tilde{\nu}_I \in \mathbb{R}^m$ and $\tilde{\nu}_E \in \mathbb{R}^p$ are the Lagrange multipliers for the equality constraints Eq. (21b). Solving this QP as-is would be expensive since it requires solving a linear system of size $n+3m+2p$. However, we can eliminate $\tilde{s}$, $\tilde{z}_I$, and $\tilde{z}_E$ using Eqs. (22b) to (22d) above, so the linear system simplifies to

$$\begin{bmatrix} P + \sigma_x I & G^\top & A^\top \\ G & -(\sigma_s^{-1} + \rho_I^{-1})I & 0 \\ A & 0 & -\rho_E^{-1}I \end{bmatrix}\begin{bmatrix} \tilde{x} \\ \tilde{\nu}_I \\ \tilde{\nu}_E \end{bmatrix} = \begin{bmatrix} \sigma_x x^k - q \\ h - s^k + \sigma_s^{-1}w_s^k + z_I^k - \rho_I^{-1}y_I^k \\ b + z_E^k - \rho_E^{-1}y_E^k \end{bmatrix}, \tag{23}$$

with the eliminated variables recoverable using

$$\tilde{s} = s^k - \sigma_s^{-1} w_s^k - \sigma_s^{-1} \tilde{\nu}_I, \tag{24a}$$

$$\tilde{z}_I = z_I^k - \rho_I^{-1} y_I^k + \rho_I^{-1} \tilde{\nu}_I, \tag{24b}$$

$$\tilde{z}_E = z_E^k - \rho_E^{-1} y_E^k + \rho_E^{-1} \tilde{\nu}_E. \tag{24c}$$

The coefficient matrix in the linear system Eq. (23) is always full rank due to the positive parameters $\sigma_x$, $\sigma_s$, $\rho_I$, and $\rho_E$ introduced through the ADMM splitting. This linear system can be solved using a direct method such as an $LDL^\top$ factorization requiring $O((n+m+p)^3)$ time, the same as OSQP using the direct method. On the other hand, for large-scale QPs, i.e., when $n + m + p$ is very large, factoring this matrix can be prohibitively expensive. In this case, we can use an indirect method to solve the reduced system

$$(P + \sigma_x I + \bar{G}^\top G + \bar{A}^\top A)\tilde{x} = \sigma_x x^k - q + \bar{G}^\top (h - s^k + \sigma_s^{-1} w_s^k + z_I^k - \rho_I^{-1} y_I^k) \tag{25}$$
$$+ \bar{A}^\top (b + z_E^k - \rho_E^{-1} y_E^k),$$

where $\bar{G} = (\sigma_s^{-1} + \rho_I^{-1})^{-1} G$ and $\bar{A} = \rho_E A$. This can be obtained by eliminating $\tilde{\nu}_I$ and $\tilde{\nu}_E$ from the linear system Eq. (23). These variables are recoverable using

$$\tilde{\nu}_I = (\sigma_s^{-1} + \rho_I^{-1})^{-1}(G\tilde{x} + s^k - \sigma_s^{-1} w_s^k - z_I^k + \rho_I^{-1} y_I^k - h), \tag{26a}$$

$$\tilde{\nu}_E = \rho_E(A\tilde{x} - z_E^k + \rho_E^{-1} y_E^k - b). \tag{26b}$$

The coefficient matrix in Eq. (25) is always positive definite, so the linear system can be solved using an iterative algorithm such as the conjugate gradient (CG) method. The linear system is of size $n$, matching the complexity of OSQP using the indirect method. In this work, we consider a supervised learning setting where we will need to compute derivatives of the solution $\tilde{x}$ with respect to the parameters $\sigma_x$, $\rho_I$, etc. While each iteration of the CG method is very fast, it can require many iterations to converge to a low-error solution. It would be very inefficient to backpropagate through all these iterations of the CG method, the main issue being the high memory cost since the entire compute graph needs to be stored and then differentiated through during the backward pass. We instead adopt the approach from Saravanos et al. (2025, Theorem 2) using differentiable optimization in order to compute these derivatives in a more efficient manner. In practice, this means we can compute the derivatives by solving a new linear system with the same coefficient matrix but different right-hand side during the backward pass.

## D  FLEXQP ALGORITHM

---

**Algorithm 1:** FlexQP

**Input**  : Initialization $x^0, s^0, z_I^0, z_E^0, w_s^0, y_I^0, y_E^0$ and parameters $\mu_I, \mu_E, \sigma_x, \sigma_s, \rho_I, \rho_E > 0$
**Output:** Solution $x^*, y_I^*, y_E^*$
**while** *termination criterion not satisfied* **do**

$\quad \tilde{x}^{k+1}, \tilde{\nu}_I^{k+1}, \tilde{\nu}_E^{k+1} \leftarrow$ Solve the linear system Eq. (23)
$\quad \tilde{s}^{k+1} = s^k - \sigma_s^{-1} w_s^k - \sigma_s^{-1} \tilde{\nu}_I^{k+1}$
$\quad \tilde{z}_I^{k+1} = z_I^k - \rho_I^{-1} y_I^k + \rho_I^{-1} \tilde{\nu}_I^{k+1}$
$\quad \tilde{z}_E^{k+1} = z_E^k - \rho_E^{-1} y_E^k + \rho_E^{-1} \tilde{\nu}_E^{k+1}$
$\quad x^{k+1} = \alpha \tilde{x}^{k+1} + (1 - \alpha) x^k$
$\quad s^{k+1} = \left( \alpha \tilde{s}^{k+1} + (1 - \alpha) s^k + \sigma_s^{-1} w_s^k \right)_+$
$\quad z_I^{k+1} = S_{\mu_I/\rho_I} \left( \alpha \tilde{z}_I^{k+1} + (1 - \alpha) z_I^k + \rho_I^{-1} y_I^k \right)$
$\quad z_E^{k+1} = S_{\mu_E/\rho_E} \left( \alpha \tilde{z}_E^{k+1} + (1 - \alpha) z_E^k + \rho_E^{-1} y_E^k \right)$
$\quad w_s^{k+1} = w_s^k + \sigma_s(\alpha \tilde{s}^{k+1} + (1 - \alpha) s^k - s^{k+1})$
$\quad y_I^{k+1} = y_I^k + \rho_I(\alpha \tilde{z}_I^{k+1} + (1 - \alpha) z_I^k - z_I^{k+1})$
$\quad y_E^{k+1} = y_E^k + \rho_E(\alpha \tilde{z}_E^{k+1} + (1 - \alpha) z_E^k - z_E^{k+1})$
**end**

---

# E    PROOF OF THEOREM 3.2

To show that a saddle point of Eq. (10) exists, it suffices to show that the relative interior of the domains of $f$ and $g$ are non-empty. This is a form of constraint qualification guaranteeing strong duality (Rockafellar, 1970, Theorem 16.4). The relative interior of $f$ is simply the feasible set $\{(x, s, z_I, z_E) : z_I = Gx + s - h, z_E = Ax - b\}$, which is non-empty (pick any $x, s$ and then let $z_I = Gx + s - h$ and $z_E = Ax - b$). Meanwhile, the relative interior of $g$ is given by the set $\{(x, s, z_I, z_E) : s > 0\}$ due to the constraint $s \geq 0$. Combining, this shows that $\mathrm{relint}(\mathrm{dom}\, f) \cap \mathrm{relint}(\mathrm{dom}\, g) \neq \emptyset$, so strong duality holds and a saddle point exists by Rockafellar (1970, Theorem 37.3).

Furthermore, as the objective $f$ and $g$ are closed, proper, and convex, by Boyd et al. (2011, §3.2), Algorithm 1 converges. Namely, we have that the ADMM primal residuals $\tilde{\boldsymbol{\zeta}}^k \to 0$ and ADMM dual residuals $\bar{\boldsymbol{\zeta}}^k \to 0$ as $k \to \infty$, where

$$\tilde{\boldsymbol{\zeta}}^k = \begin{bmatrix} \tilde{\zeta}_x^k \\ \tilde{\zeta}_s^k \\ \tilde{\zeta}_I^k \\ \tilde{\zeta}_E^k \end{bmatrix} = \begin{bmatrix} \tilde{x}^k - x^k \\ \tilde{s}^k - s^k \\ \tilde{z}_I^k - z_I^k \\ \tilde{z}_E^k - z_E^k \end{bmatrix}, \quad \bar{\boldsymbol{\zeta}}^k = \begin{bmatrix} \bar{\zeta}_x^k \\ \bar{\zeta}_s^k \\ \bar{\zeta}_I^k \\ \bar{\zeta}_E^k \end{bmatrix} = \begin{bmatrix} x^{k-1} - x^k \\ s^{k-1} - s^k \\ z_I^{k-1} - z_I^k \\ z_E^{k-1} - z_E^k \end{bmatrix}. \tag{27}$$

Furthermore, we also have that iterates $\tilde{\boldsymbol{x}}^k \to \hat{\tilde{\boldsymbol{x}}}$, $\boldsymbol{x}^k \to \hat{\boldsymbol{x}}$, and $\boldsymbol{y}^k \to \hat{\boldsymbol{y}}$ as $k \to \infty$.

# F    PROOF OF THEOREM 3.3

The proof follows from the definition of the soft thresholding operator. First, we consider the $z_I$ and $z_E$ updates as well as the dual variable updates for $y_I$ and $y_E$ from Eq. (9). Assume w.l.o.g. that $\alpha = 1$. These updates have the general form:

$$z^{k+1} = S_{\mu/\rho}\left(\tilde{z}^{k+1} + y^k/\rho\right), \tag{28a}$$

$$y^{k+1} = y^k + \rho(\tilde{z}^{k+1} - z^{k+1}). \tag{28b}$$

Now, there are three cases the consider based on the output of the soft thresholding operation:

1. **Positive constraint violation:** If $\tilde{z}^{k+1} + y^k/\rho > \mu/\rho$, then $z^{k+1} = \tilde{z}^{k+1} + y^k/\rho - \mu/\rho$. Substituting into Eq. (28b) yields $y^{k+1} = \mu$.

2. **No constraint violation:** If $|\tilde{z}^{k+1} + y^k/\rho| \leq \mu/\rho$, then $z^{k+1} = 0$. This further implies $\rho|\tilde{z}^{k+1} + y^k/\rho| \leq \mu$ and by Eq. (28b) this implies $|y^{k+1}| \leq \mu$.

3. **Negative constraint violation:** If $\tilde{z}^{k+1} + y^k/\rho < -\mu/\rho$, then $z^{k+1} = \tilde{z}^{k+1} + y^k/\rho + \mu/\rho$. Substituting into Eq. (28b) yields $y^{k+1} = -\mu$.

Combining these three cases, we have that $|y^{k+1}| \leq \mu$, for any $\tilde{z}^{k+1}, y^k$ and therefore $|\hat{y}| \leq \mu$. This proves the first and last statement of the theorem. Applying Theorem 3.1 shows the second statement.

# G    DEEP FLEXQP POLICY PARAMETERIZATION

The residuals for Eq. (7) are given by

$$\zeta_{\text{dual}}^k = Px^k + q + G^\top y_I^k + A^\top y_E^k, \tag{29a}$$

$$\zeta_I^k = Gx^k + s^k - h - z_I^k, \tag{29b}$$

$$\zeta_E^k = Ax^k - b - z_E^k. \tag{29c}$$

The ADMM residuals for Eq. (8) are defined in Eq. (27). We ignore the residuals corresponding to $x$ since it is unconstrained in the second ADMM block (so the primal residual is not very meaningful) and we have already captured the optimality through Eq. (29).

The policy $\pi_I : \mathbb{R}^{10} \to \mathbb{R}^3_+$ is given by

$$\mu_I, \sigma_s, \rho_I = \pi_I(s, z_I, w_s, y_I, \|\zeta_{\text{dual}}\|_\infty, \zeta_I, \bar{\zeta}_s, \bar{\zeta}_I, \tilde{\zeta}_s, \tilde{\zeta}_I), \tag{30}$$

where we have dropped the indices by constraint $i$ and iteration $k$ for clarity.

The policy $\pi_E : \mathbb{R}^6 \to \mathbb{R}^2_+$ is given by

$$\mu_E, \rho_E = \pi_E(z_E, y_E, \|\zeta_{\text{dual}}\|_\infty, \zeta_E, \bar{\zeta}_E, \tilde{\zeta}_E). \tag{31}$$

The policy $\pi_\alpha : \mathbb{R}^9 \to (0, 2)$ is given by

$$\alpha = \pi_\alpha(\|\zeta_{\text{dual}}\|, \|\zeta_I\|, \|\zeta_E\|, \|\bar{\zeta}_s\|, \|\bar{\zeta}_I\|, \|\bar{\zeta}_E\|, \|\tilde{\zeta}_s\|, \|\tilde{\zeta}_I\|, \|\tilde{\zeta}_E\|), \tag{32}$$

where the norm used is the infinity norm.

We consider two policy parameterizations: multilayer perceptrons (MLPs) and LSTMs (see Appendix N for experimental comparison). Following Saravanos et al. (2025), MLP policies are small networks with two hidden layers of sizes [32, 32]. LSTM policies use a hidden size of 32 followed by an MLP with hidden layers [32, 32] for prediction. We use sigmoid for all activation functions, which we find are much more stable than ReLU activations, most likely due to the autoregressive nature of deep unfolding. Computationally, we log-scale any small positive inputs like the infinity norms of the residuals. Following Ichnowski et al. (2021), we also predict log-transformed values $\log \mu_I$, $\log \rho_E$, etc. and then apply an exponential function so that it is easier to predict parameters across a wide scale of values. We then clamp the parameters (besides $\alpha$) to the range $[10^{-6}, 10^6]$; $\alpha \in (0, 2)$ is enforced using a scaled sigmoid function.

## H   FURTHER DETAILS ON QP PROBLEM CLASSES

A summary of the problem sizes and training parameters of the different classes is presented in Table 1. For the small- to medium-scale QPs, we train all models for 500 epochs and evaluate using 1000 test samples. More training samples are used for random QPs following the setup by Saravanos et al. (2025) since the shared structure between these QPs is less clear and therefore harder to learn. As described in the main text, for the large-scale problems, we fine-tune the models trained on the smaller scale problems on 100 large-scale problems for 5 epochs, and test on 100 new problems.

Table 1: QP problem sizes and number of samples used for training.

| Problem Class | $n$ | $m$ | $p$ | Training Samples |
|---|---|---|---|---|
| Random QPs | 50 | 40 | 0 | 2000 |
| Random QPs with Equalities | 50 | 25 | 20 | 2000 |
| Portfolio Optimization | 275 | 250 | 26 | 500 |
| Support Vector Machine | 210 | 400 | 0 | 500 |
| LASSO | 510 | 10 | 500 | 500 |
| Huber Fitting | 310 | 200 | 100 | 500 |
| Random Linear OCPs | 128 | 256 | 88 | 500 |
| Double Integrator | 62 | 124 | 42 | 500 |
| Oscillating Masses | 162 | 324 | 132 | 500 |
| Portfolio Optimization (Large-Scale) | 10100 | 10000 | 101 | 100 |
| Support Vector Machine (Large-Scale) | 10100 | 20000 | 0 | 100 |
| Car with Obstacles (SQP) | 253 | 455 | 153 | 500 |
| Quadrotor (SQP) | 812 | 400 | 612 | 500 |
| Car Safety Filter (SQP) | 253 | 50 | 153 | 500 |

### H.1   RANDOM QPS

The first type of problems we study are random QPs of the form Eq. (3). These are helpful as we can freely adjust the number of constraints as well as the sparsity of the problem directly in order to benchmark the optimizers under different operating conditions.

**Problem Instances:** We adopt the problem generation procedure from Saravanos et al. (2025), where $P = M^\top M + \alpha I$ with $\alpha = 1$ and all elements of $M$, $q$, $G$, and $A$ are standard normal distributed, i.e., each element $M_{ij}, q_i, G_{ij}, A_{ij} \sim \mathcal{N}(0,1)$. The vectors $h$ and $b$ are generated using $h = G\xi$ and $b = A\zeta$ with $\xi, \zeta$ standard normal vectors.

We consider two classes of random QPs. The first class, **Random QPs**, contains only inequality constraints generated by setting the problem dimensions as $n = 50$, $m = 40$, and $p = 0$. The second class, **Random QPs with Equalities** contains a mix of inequality and equality constraints, and is generated using $n = 50$, $m = 25$, and $p = 20$.

## H.2   Portfolio Optimization

**Portfolio Optimization** is a foundational problem in finance where the goal is to maximize the risk-adjusted return of a group of assets (Markowitz, 1952; Boyd et al., 2013; 2017). This can be represented as the following QP (Boyd & Vandenberghe, 2004; Stellato et al., 2020):

$$\max_x \quad \mu^\top x - \gamma(x^\top \Sigma x), \tag{33a}$$

$$\text{subject to} \quad \mathbf{1}^\top x = 1, \tag{33b}$$

$$x \geq 0, \tag{33c}$$

where $x \in \mathbb{R}^n$ is the portfolio, $\mu \in \mathbb{R}^n$ is the expected returns, $\gamma > 0$ is the risk aversion parameter, and $\Sigma \in \mathbb{S}_+^n$ is the risk model covariance.

**QP Representation:** We assume that $\Sigma = FF^\top + D$ where $F \in \mathbb{R}^{n \times k}$ is the rank-$k$ factor loading matrix with $k < n$ and $D \in \mathbb{R}^{n \times n}$ is the diagonal matrix specifying the asset-specific risk. Using this assumption, the optimization problem can be converted into a more efficient QP representation:

$$\min_{x,y} \quad x^\top D x + y^\top y - \gamma^{-1} \mu^\top x, \tag{34a}$$

$$\text{subject to} \quad y = F^\top x, \tag{34b}$$

$$\mathbf{1}^\top x = 1, \tag{34c}$$

$$x \geq 0. \tag{34d}$$

This new QP has $n + k$ decision variables, $k + 1$ equality constraints, and $n$ inequality constraints.

**Problem Instances:** For the medium-scale problems, we use the problem generation described in Saravanos et al. (2025), setting $n = 250$, $k = 25$, and $\gamma = 1.0$. The large-scale problems are generated using $n = 10000$ assets, $k = 100$ factors, and $\gamma = 1.0$. The expected returns $\mu$ are sampled using $\mu_i \sim \mathcal{N}(0,1)$. The factor loading matrix $F$ has 50% non-zero elements sampled through $F_{ij} \sim \mathcal{N}(0,1)$. The diagonal elements of $D$ are generated uniformly as $D_{ii} \sim \mathcal{U}(0, \sqrt{k})$.

## H.3   Support Vector Machines

**Support Vector Machines** (SVMs) are a classical machine learning method where the goal is to find a linear classifier that best separates two sets of points (Cortes & Vapnik, 1995):

$$\min_x x^\top x + \lambda \sum_{i=1}^m \max(0, b_i a_i^\top x + 1), \tag{35}$$

where $\lambda > 0$, $b_i \in \{-1, 1\}$ is the label, and $a_i \in \mathbb{R}^n$ is the set of features for point $i$.

**QP representation:** The SVM problem Eq. (35) can be converted into an equivalent QP representation (Stellato et al., 2020):

$$\min_{x,t} \quad x^\top x + \lambda \mathbf{1}^\top t, \tag{36a}$$

$$\text{subject to} \quad t \geq \text{diag}(b)Ax + 1, \tag{36b}$$

$$t \geq 0. \tag{36c}$$

This QP has $n + m$ decision variables and $2m$ inequality constraints.

**Problem Instances:** We generate medium-scale problems using the rules from Stellato et al. (2020) with $n = 10$ features, $m = 200$ data points, and $\lambda = 1$. Large-scale problems are generated using $n = 100$ features, $m = 10000$ data points, and $\lambda = 1$. The labels $b$ are chosen using

$$b_i = \begin{cases} +1 & \text{if } i \leq m/2, \\ -1 & \text{otherwise,} \end{cases} \tag{37}$$

and the elements of $A$ are chosen such that

$$A_{ij} \sim \begin{cases} \mathcal{N}(+1/n, 1/n) & \text{if } i \leq m/2, \\ \mathcal{N}(-1/n, 1/n) & \text{otherwise.} \end{cases} \tag{38}$$

## H.4 LASSO

**LASSO** (least absolute shrinkage and selection operator) is a fundamental problem in statistics and machine learning (Tibshirani, 1996; Candes et al., 2008). The objective is to select sparse coefficients of a linear model that best match the given observations:

$$\min_x \|Ax - b\|_2^2 + \lambda \|x\|_1, \tag{39}$$

where $x \in \mathbb{R}^n$, $A \in \mathbb{R}^{m \times n}$ is the data matrix, $b \in \mathbb{R}^m$ are the observations, and $\lambda > 0$ is the weighting parameter.

**QP Representation:** LASSO can be represented as a QP by introducing two extra decision variables $y \in \mathbb{R}^m$ and $t \in \mathbb{R}^n$ which help simplify the objective (Stellato et al., 2020):

$$\min_{x,y,t} \quad y^\top y + \lambda \mathbf{1}^\top t, \tag{40a}$$

$$\text{subject to} \quad y = Ax - b, \tag{40b}$$

$$-t \leq x \leq t. \tag{40c}$$

**Problem Instances:** We use the data generation procedure from (Stellato et al., 2020), where $A$ has 15% non-zero normally-distributed elements $A_{ij} \sim \mathcal{N}(0, 1)$ and $b$ is generated through $b = Av + \epsilon$ with

$$v_i \sim \begin{cases} 0 & \text{with probability } p = 0.5, \\ \mathcal{N}(0, 1/n) & \text{otherwise,} \end{cases} \tag{41}$$

and $\epsilon_i \sim \mathcal{N}(0, 1)$. The parameter $\lambda$ is chosen as $\lambda = (1/5)\|A^\top b\|_\infty$.

## H.5 HUBER FITTING

**Huber Fitting** is a robust least squares problem where the goal is to perform a linear regression with the assumption that outliers are present in the data (Huber, 1964; 1981):

$$\min_x \sum_{i=1}^m \phi_{\text{hub}}(a_i^\top x - b_i), \tag{42}$$

where the penalty function $\phi_{\text{hub}}$ penalizes the residuals linearly when they are large and quadratically when they are small:

$$\phi_{\text{hub}}(u) = \begin{cases} u^2 & \text{if } |u| \leq \delta, \\ \delta(2|u| - \delta) & \text{if } |u| > \delta, \end{cases} \tag{43}$$

with $\delta > 0$ representing the slope of the linear term.

**QP Representation:** This robust least squares problem can be represented in the following QP form (Stellato et al., 2020):

$$\min_{x,u,r,s} \quad u^\top u + 2\delta \mathbf{1}^\top (r + s), \tag{44a}$$

$$\text{subject to} \quad Ax - b - u = r - s, \tag{44b}$$

$$r, s \geq 0. \tag{44c}$$

This QP has $n + 3m$ decision variables, $2m$ inequalities, and $m$ equalities.

**Problem Instances:** We follow Stellato et al. (2020) and generate $A$ with 15% nonzero elements with $A_{ij} \sim \mathcal{N}(0, 1)$ and set $b = Av + \epsilon$ where

$$\epsilon_i = \begin{cases} \mathcal{N}(0, 1/4) & \text{with probability } p = 0.95, \\ \mathcal{U}(0, 10) & \text{otherwise.} \end{cases} \tag{45}$$

We let $\delta = 1$ and choose the problem dimensions as $n = 10$ features and $m = 10n = 100$ datapoints.

## H.6 LINEAR OPTIMAL CONTROL

The goal in linear optimal control is to stabilize the system to the origin subject to dynamical constraints as well as polyhedral constraints on the states and controls. This results in QPs of the form

$$\min_{x,u} \quad \sum_{t=0}^{T-1} x_t^\top Q x_t + u_t^\top R u_t + x_T^\top Q_T x_T, \tag{46a}$$

$$\text{subject to} \quad x_{t+1} = A_d x_t + B_d u_t, \tag{46b}$$

$$A_u u_t \leq b_u, \tag{46c}$$

$$A_x x_t \leq b_x, \tag{46d}$$

$$x_0 = \bar{x}_0, \tag{46e}$$

where $T > 0$ is the time horizon, $Q \in \mathbb{S}_+^{n_x}$ is the running state cost matrix, $R \in \mathbb{S}_{++}^{n_u}$ is the control cost matrix, $Q_T \in \mathbb{S}_+^{n_x}$ is the terminal state cost matrix, $A_d \in \mathbb{R}^{n_x \times n_x}$ and $B_d \in \mathbb{R}^{n_x \times n_u}$ define the dynamics of the system, $A_u \in \mathbb{R}^{m_u \times n_u}$ and $b_u \in \mathbb{R}^{m_u}$ define the input constraints, $A_x \in \mathbb{R}^{m_x \times n_x}$ and $b_x \in \mathbb{R}^{m_x}$ define the state constraints, and $\bar{x}_0 \in \mathbb{R}^{n_x}$ is the initial condition.

We study three classes of linear optimal control problems (OCPs). The first, **Random Linear OCPs**, consists of randomly generated stabilizable dynamics along with random costs, constraints, and initial conditions. The second and third classes, **Double Integrator** and **Oscillating Masses**, are adapted from Chen et al. (2022a) and contain dynamics with true physical interpretations. The randomness in these problems is given by sampling varying initial conditions for the systems as in Saravanos et al. (2025).

### H.6.1 RANDOM LINEAR OCPS

We use the problem generation procedure similar to that in Stellato et al. (2020). We set the state dimension $n_x = 8$ and $n_u = n_x/2 = 4$. The dynamics are generated by $A_d = X^{-1} A X$, where $A = \text{diag}(a) \in \mathbb{R}^{n_x \times n_x}$ such that $a_i \sim \mathcal{U}(-1, 1)$ and $X \in \mathbb{R}^{n_x \times n_x}$ with elements generated by $X_{ij} \sim \mathcal{N}(0, 1)$, and $B_d \in \mathbb{R}^{n_x \times n_u}$ with $(B_d)_{ij} \sim \mathcal{N}(0, 1)$.

The running state cost $Q \in \mathbb{S}_+^{n_x}$ is generated by $Q = \text{diag}(q)$ where each element of the sparse vector $q$ is generated by

$$q_i \sim \begin{cases} \mathcal{U}(0, 10) & \text{with probability } p = 0.7, \\ 0 & \text{otherwise,} \end{cases} \tag{47}$$

so that $q$ has 70% nonzero values. We fix the control cost $R = 0.1 I_u$ and the terminal cost $Q_T$ is determined by solving the discrete algebraic Riccati for the optimal cost of a linear quadratic regulator applied to $A, B, Q$, and $R$. The state and control constraints are generated by

$$A_x = \begin{bmatrix} I_x \\ -I_x \end{bmatrix}, \quad b_x = \begin{bmatrix} x^{\text{bound}} \\ -x^{\text{bound}} \end{bmatrix}, \quad \text{where } x_i^{\text{bound}} \sim \mathcal{U}(1, 2), \tag{48a}$$

$$A_u = \begin{bmatrix} I_u \\ -I_u \end{bmatrix}, \quad b_u = \begin{bmatrix} u^{\text{bound}} \\ -u^{\text{bound}} \end{bmatrix}, \quad \text{where } u_i^{\text{bound}} \sim \mathcal{U}(0, 0.1). \tag{48b}$$

Note that we use $I_x$ and $I_u$ as a shorthand for $I_{n_x}$ and $I_{n_u}$. Finally, we sample the initial state from $\bar{x}_0 \sim \mathcal{U}(-0.5 x^{\text{bound}}, 0.5 x^{\text{bound}})$.

### H.6.2 DOUBLE INTEGRATOR

For the double integrator, adapted from Chen et al. (2022a), we have $n_x = 2$, $n_u = 1$, and $T = 20$ timesteps. The dynamics are fixed with

$$A_d = \begin{bmatrix} 1 & 1 \\ 0 & 1 \end{bmatrix}, B_d = \begin{bmatrix} 0.5 \\ 0.1 \end{bmatrix}. \tag{49}$$

We use cost matrices $Q = Q_T = I_x$ and $R = 1.0$. The state and control constraints are given by

$$A_x = \begin{bmatrix} I_x \\ -I_x \end{bmatrix}, b_x = \begin{bmatrix} 5 \\ 1 \\ 5 \\ 1 \end{bmatrix}, A_u = \begin{bmatrix} 1 \\ -1 \end{bmatrix}, b_u = \begin{bmatrix} 0.1 \\ 0.1 \end{bmatrix}. \tag{50}$$

The initial state is sampled from $\bar{x}_0 \sim \mathcal{U}\left(\begin{bmatrix} -1 \\ -0.3 \end{bmatrix}, \begin{bmatrix} 1 \\ 0.3 \end{bmatrix}\right)$.

### H.6.3 OSCILLATING MASSES

For the oscillating masses problem, we have $n_x = 12$, $n_u = 3$, $T = 10$. For this problem, the discrete-time dynamics matrices $A_d$ and $B_d$ are obtained through the Euler discretization of the continuous-time dynamics of the oscillating masses system, namely

$$A_d = I_x + A_c \Delta t, \quad B_d = B_c \Delta t, \tag{51}$$

where $\Delta t = 0.5$. The matrices $A_c \in \mathbb{R}^{n_x \times n_x}$ and $B_c \in \mathbb{R}^{n_x \times n_u}$ define the continuous-time dynamics and are given by

$$A_c = \begin{bmatrix} 0_{6\times6} & I_6 \\ aI_6 + c(L_6 + L_6^\top) & bI_6 + d(L_6 + L_6^\top) \end{bmatrix}, B_c = \begin{bmatrix} 0_{6\times3} \\ F \end{bmatrix}, \tag{52}$$

where $c = 1$, $d = 0.1$, $a = -2c$, $b = 2$, $0_{m\times n}$ is the zero matrix in $\mathbb{R}^{m \times n}$, $L_n$ is the lower shift matrix in $\mathbb{R}^{n \times n}$, and $F = \begin{bmatrix} e_1 & -e_1 & e_2 & e_3 & -e_2 & e_3 \end{bmatrix}^\top$, where $e_1, e_2$, and $e_3$ are the standard basis vectors in $\mathbb{R}^3$. We use cost matrices $Q = Q_T = I_x$ and $R = I_u$. The state and control constraints are given by

$$A_x = \begin{bmatrix} I_x \\ -I_x \end{bmatrix}, \quad b_x = 4 \cdot \mathbf{1}_x, \quad A_u = \begin{bmatrix} I_u \\ -I_u \end{bmatrix}, \quad b_u = 0.5 \cdot \mathbf{1}_u. \tag{53}$$

Finally, we sample the initial state from $\bar{x}_0 \sim \mathcal{U}(-\mathbf{1}_x, \mathbf{1}_x)$.

# I   NONCONVEX NONLINEAR PROGRAMMING USING SQP

## I.1   NONLINEAR OPTIMAL CONTROL

We consider nonlinear constrained optimal control problems of the following form:

$$\underset{x,u}{\text{minimize}} \quad \sum_{t=0}^{T-1} \ell(x_t, u_t) + \phi(x_T), \tag{54a}$$

$$\text{subject to} \quad x_{t+1} = F(x_t, u_t), \quad \forall t = 0, \dots, T-1, \tag{54b}$$

$$x_0 = \bar{x}_0, \tag{54c}$$

$$h(x_t) \leq 0, \quad \forall t = 0, \dots, T, \tag{54d}$$

$$g(u_t) \leq 0, \quad \forall t = 0, \dots, T-1, \tag{54e}$$

where $x_t \in \mathbb{R}^n$ and $u_t \in \mathbb{R}^m$ are the states and controls, respectively. The function $\ell : \mathbb{R}^n \times \mathbb{R}^m \to \mathbb{R}$ is the running cost and $\phi : \mathbb{R}^n \to \mathbb{R}$ is the terminal cost. The time horizon is $T > 0$ and $\bar{x}_0 \in \mathbb{R}^n$ is the initial condition. The problem formulation in Eq. (54) includes state and control constraints represented by the functions $h(x_t)$ and $g(u_t)$. In the next two subsections, we provide the specific nonlinear optimal control examples for the cases of the Dubins vehicle and quadrotor.

### I.1.1   DUBINS VEHICLE

The Dubins vehicle is a dynamics model with a state $x = (p_x, p_y, \theta) \in \mathbb{R}^3$, where $p_x$ and $p_y$ are the vehicle's position in the Cartesian plane and $\theta$ is its orientation. We use the unicycle formulation of the continuous-time Dubins vehicle dynamics $\dot{x} = f(x, u)$ adapted from Siciliano et al. (2009), where the control is given by $u = (v, \omega) \in \mathbb{R}^2$. Here, $v$ is the forward velocity of the vehicle and $\omega$ is the steering velocity of the vehicle.

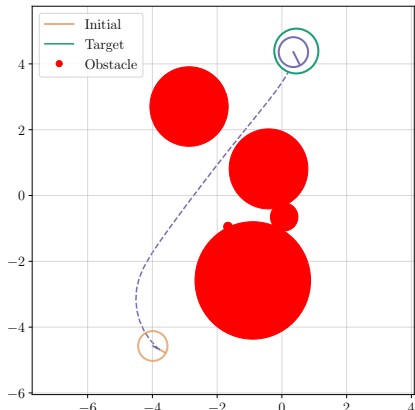

Figure 8: Visualization of a sample Dubins vehicle task. The goal is to reach the target state while avoiding obstacles and respecting the dynamics and input constraints.

We formulate a nonlinear optimal control problem following Eq. (54). We discretize the continuous-time dynamics using the Euler discretization $x_{t+1} = F(x_t, u_t) = x_t + f(x_t, u_t)\Delta t$ and use quadratic costs $\ell(x, u) = x^\top Q x + u^\top R u$ and $\phi(x) = x^\top Q_T x$, where $Q = \text{diag}(1.0, 1.0, 0.1)$, $R = 0.1 \cdot I$, and $Q_T = 100 \cdot Q$. The initial and target state, $x_0$ and $x_{\text{target}}$, are sampled uniformly from $\mathcal{U}(-\bar{x}, \bar{x})$, where $\bar{x} = (5.0, 5.0, \pi)$. The discretization of the dynamics uses $\Delta t = 0.033$ and the time horizon for trajectory optimization is $T = 50$ timesteps. We generate 5 circular obstacles with the form

$$g_i(x_t) = r_i^2 - \|x_t - c_i\|_2^2 \leq 0, \quad \forall t = 0, \dots, T, \tag{55}$$

where the centers $c_i \in \mathbb{R}^2$ are sampled uniformly at random in the region between the vehicle's initial and target positions, and the radii $r_i > 0$ are sampled uniformly from $\mathcal{U}(r_{\min}, r_{\max})$ with $r_{\min} = 0.01 \cdot \|x_{\text{target}} - x_0\|_2$ and $r_{\max} = 0.2 \cdot \|x_{\text{target}} - x_0\|_2$. The controls are constrained by

$v \in [-10, 10]$ and $\omega \in [-5, 5]$. This leads to a nonlinear optimization problem with 253 variables, 455 inequality constraints, and 153 equality constraints.

For generating the QP training data, we generate 500 QP subproblems by solving randomly generated Dubins vehicle problems with SQP using OSQP as the QP solver. For evaluation, we generate 100 random control problems and solve them using SQP with OSQP or SQP with Deep FlexQP. Each algorithm is allowed 50 SQP iterations and runs until the infinity norm of the SQP residuals falls below an absolute tolerance of $\varepsilon = 10^{-2}$. Furthermore, each QP solver runs until convergence of $10^{-3}$ is reached, with a max budget of 10 seconds and an unlimited number of iterations.

### I.1.2 QUADROTOR

We use the continuous-time quadrotor dynamics model $\dot{x} = f(x, u)$ from Sabatino (2015). The model consists of the state $x \in \mathbb{R}^{12}$ that includes the linear positions, angles, linear velocities, and angular velocities. The system is actuated by four controls, the collective thrust $F$ and three torques, given by $u = (F, \tau_x, \tau_y, \tau_z) \in \mathbb{R}^4$.

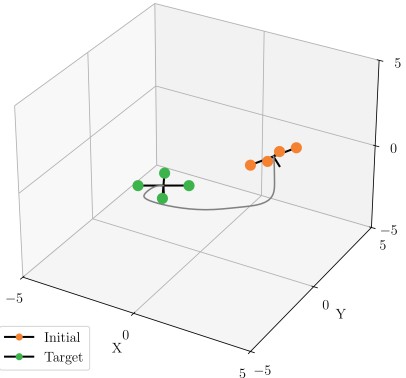

Figure 9: Visualization of a sample quadrotor task. The goal is to reach the target state from the initial state subject to dynamical constraints and input constraints.

Using this model, we formulate nonlinear optimal control problems as in Eq. (54). We discretize the dynamics through the Euler discretization $x_{t+1} = F(x_t, u_t) = x_t + f(x_t, u_t)\Delta t$. The cost in Eq. (54) is defined by the quadratic cost $\ell(x, u) = x^\top Q x + u^\top R u$ and $\phi(x) = x^\top Q_T x$, where the cost matrices are given by

$$Q = \mathrm{diag}(1.0, 1.0, 1.0, 0.1, 0.1, 0.1, 1.0, 1.0, 1.0, 0.1, 0.1, 0.1),$$

along with $R = 0.01 \cdot I$ and $Q_T = 1000 \cdot Q$. The initial and target state are sampled uniformly from $\mathcal{U}(-\bar{x}, \bar{x})$ where $\bar{x} = (5, 5, 5, 1, 1, 1, \pi, \pi/2, \pi, 1, 1, 1)$. The discretization of the dynamics uses $\Delta t = 0.05$ and the time horizon for trajectory optimization is $T = 50$ timesteps. The controls are constrained in the ranges $F \in [0, 20]$ and $\tau_x, \tau_y, \tau_z \in [-10, 10]$. We use $m = 1.0$ kg for the mass of the quadrotor, $I_x = I_z = I_y = 1.0$ for its moments, and $g = 9.81$ for the acceleration due to gravity. This leads to a nonlinear optimization problem with 812 variables, 400 inequality constraints, and 612 equality constraints.

For generating the QP training data, we generate 500 QP subproblems by solving randomly generated quadrotor problems with SQP using OSQP as the QP solver. For evaluation, we generate 100 random quadrotor problems and solve them using SQP with OSQP or SQP with Deep FlexQP. Similar to the Dubins vehicle, each algorithm is allowed 50 SQP iterations and success in Fig. 7 (left) is achieved when the infinity norm of the SQP residuals falls below the absolute convergence tolerance $\varepsilon = 10^{-2}$. Each QP solver runs until convergence of $10^{-3}$ is reached, with a max budget of 10 seconds and an unlimited number of iterations.

## I.2 Nonlinear Predictive Safety Filters

Finally, we apply our proposed approach to accelerate a predictive safety filter for nonlinear model predictive control. These methods are based on control barrier functions (CBFs) (Ames et al., 2019) and filter a reference control $u^{\text{ref}}$ so that it better respects safety constraints (Wabersich & Zeilinger, 2021). The following optimization is solved at every MPC step:

$$\underset{x,u}{\text{minimize}} \quad \sum_{t=0}^{T-1} \left\| u_t - u_t^{\text{ref}} \right\|_2^2, \tag{56a}$$

$$\text{subject to} \quad x_{t+1} = F(x_t, u_t), \quad \forall t = 0, \dots, T-1, \tag{56b}$$

$$x_0 = \bar{x}_0 \tag{56c}$$

$$(1 - \beta)h(x_t) - h(x_{t+1}) \leq 0, \quad \forall t = 0, \dots, T-1, \tag{56d}$$

$$g(u_t) \leq 0, \quad \forall t = 0, \dots, T-1, \tag{56e}$$

where $\beta \in (0, 1)$ is a parameter controlling the strength of the CBF constraint. This differs from Eq. (54) because the discrete CBF constraint is defined between two consecutive states $x_t$ and $x_{t+1}$ rather than assuming the state constraints are separable across time. Our method improves upon the Shield-MPPI method proposed by Yin et al. (2023) because our optimization explicitly incorporates the dynamics and input constraints while also minimizing the discrepancy from the reference control trajectory. Furthermore, using our accelerated Deep FlexQP ensures that the optimization can be run fast enough for real-time control. We use a version of Deep FlexQP with performance guarantees from minimizing the generalization bound loss (see Appendix M).

### I.2.1 Shield-MPPI

The method by Yin et al. (2023) approximately solves a nonlinear optimization at every MPC step to generate safe controls given a trajectory from a high-level planner such as a model predictive path integral (MPPI) controller. While the main motivation behind this approach is that it is computationally fast, unfortunately, there are a few flaws with the method because it has no real guarantees of safety and the MPPI trajectory is only used to warm-start this second optimization. The main bottleneck preventing us from solving a more complex optimization in real-time is the solver speed. Therefore, this is an application where accelerating optimizers using deep unfolding can shine.

### I.2.2 Randomized Problem Scenarios

We use the same Dubins vehicle model as in Appendix I.1.1. 100 random scenarios are generated by first sampling a random initial and target state uniformly from $\mathcal{U}((-5, -5, -\pi), (5, 5, \pi))$. An obstacle is randomly sampled so its position falls between the initial and target state with a random radius $r$ depending on the distance between the initial and target state: $r \sim \mathcal{U}(0.01, 2) * \max(|p_x^{\text{target}} - p_x^{\text{init}}|, |p_y^{\text{target}} - p_y^{\text{init}}|)$. The controls are constrained in the ranges $v \in [-10, 10]$ and $\omega \in [-5, 5]$, enforced by clamping for Shield-MPPI and through the constraints of Eq. (56) for our SQP-based method. The reference trajectory at every MPC step is given by running an MPPI controller that samples 10000 trajectories with a look-ahead horizon of 50 timesteps; with a dynamics discretization of $\Delta t = 0.05$, this corresponds to a planning horizon of 2.5 seconds ahead. The system experiences zero-mean Gaussian disturbances in its state at every MPC step with standard deviation $(0.05, 0.05, 0.01)$. Shield-MPPI is allowed to run up to 5 Gauss-Newton iterations per MPC step, while our SQP safety filter is allowed to run up to 5 SQP iterations per MPC step. These thresholds were determined by estimating the max number of iterations that would still allow for real-time control of the system. Collisions in Fig. 7 (right) are counted if the state violates the CBF constraint (i.e., intersects the obstacle). Successes are counted if the vehicle reaches within a 0.1 radius of the target state. Fig. 10 shows the problem setup and sample trajectories of our approach compared with Shield-MPPI.

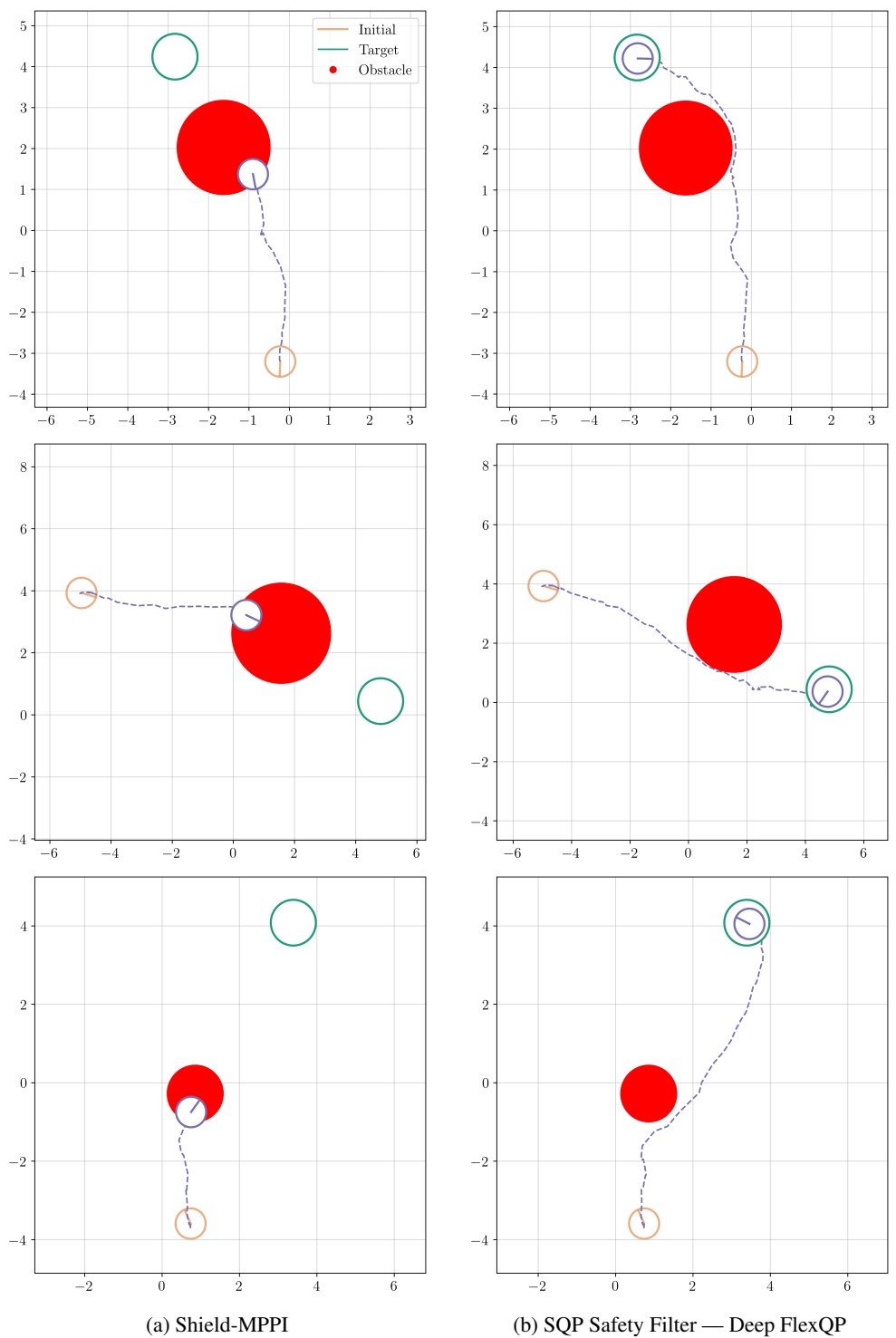

(a) Shield-MPPI        (b) SQP Safety Filter — Deep FlexQP

Figure 10: Sample trajectories comparing safety filter approaches on a nonlinear car system. The vehicle receives disturbances in the positions and orientation at every step. Our approach more effectively recovers from unsafe scenarios by better accounting for dynamic feasibility and constraints.

# J  PERFORMANCE COMPARISONS — SMALL- TO MEDIUM-SCALE QPS

The vanilla and learned optimizers from Section 5.1 are benchmarked on 1000 test QPs from each problem class with the results summarized in Fig. 11. The best version of OSQP is found using a hyperparameter search over the following configurations: fixed parameters for all iterations, adaptive penalty parameters using the OSQP rule, or adaptive penalty parameters using the ADMM rule. Similarly, the best version of FlexQP is found using a hyperparameter search over the following configurations: fixed parameters for all iterations, adaptive penalty parameters using an OSQP-like rule, or adaptive penalty parameters using the ADMM rule. Problems are considered solved when the infinity norm of the QP residuals reaches below an absolute tolerance of $\varepsilon = 10^{-3}$. Optimizers are run with no limit on the number of iterations until a timeout of 1 second (1000 ms) is reached. Timings are compared using the normalized shifted geometric mean, which is the factor at which a specific solver is slower than the fastest one (Mittelmann, 2010). We also compare the average number of iterations required to converge as well as the number of coefficient matrix factorizations required to converge to get a sense of where the optimizers are spending the most time. All methods use the direct method to solve their respective linear systems (i.e., equality-constrained QPs) at every iteration, and the factorization from the previous iteration is reused if the parameters have not changed by more than a factor of 5x following the heuristic used by OSQP (Stellato et al., 2020).

**Key Takeaways:**

1. Deep FlexQP and Deep OSQP — Improved solve QPs 2-5x faster than OSQP.

2. Deep FlexQP and Deep OSQP — Improved require upwards of 10x less iterations to converge than OSQP and require a comparable amount of matrix factorizations.

3. Deep OSQP — RLQP Parameterization struggles on problems with optimal control structure. This is not observed with Deep OSQP — Improved. Learning the ADMM relaxation parameter $\alpha$ seems to be crucial for these problems.

It is important to note that these results hold only when the direct method is used to solve the linear system. When using an indirect method such as the conjugate gradient (CG) method, converging in fewer iterations is actually a major benefit as each iteration across all the optimizers have roughly the same computational complexity (see Appendix C for discussion and Appendix K for results with the indirect method).

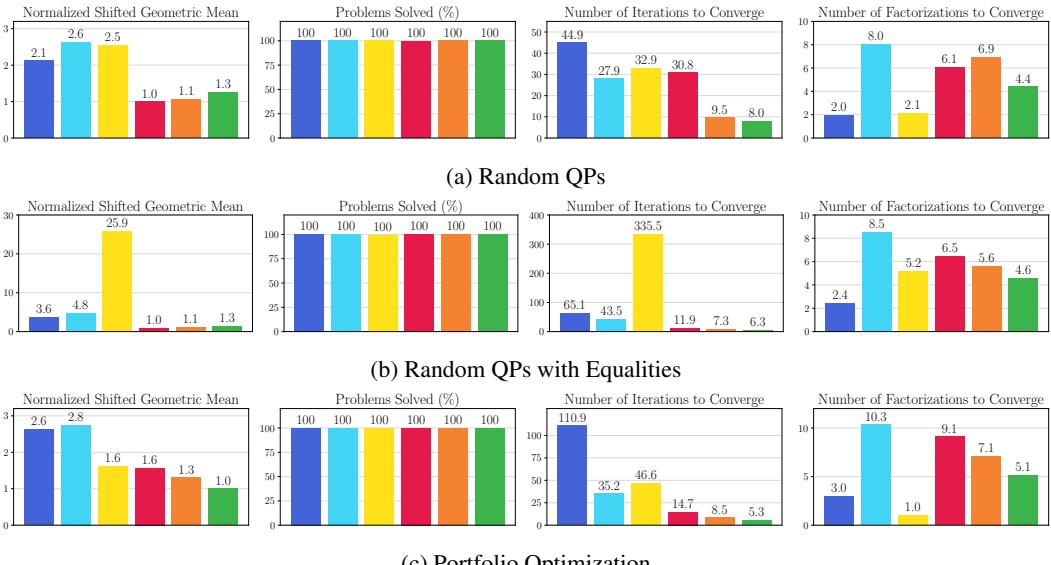

(a) Random QPs

(b) Random QPs with Equalities

(c) Portfolio Optimization

Figure 11: Performance comparison of vanilla vs. learned optimizers. Legend: OSQP, FlexQP (Ours), Deep OSQP, Deep OSQP — RLQP Parameterization, Deep OSQP — Improved, Deep FlexQP (Ours).

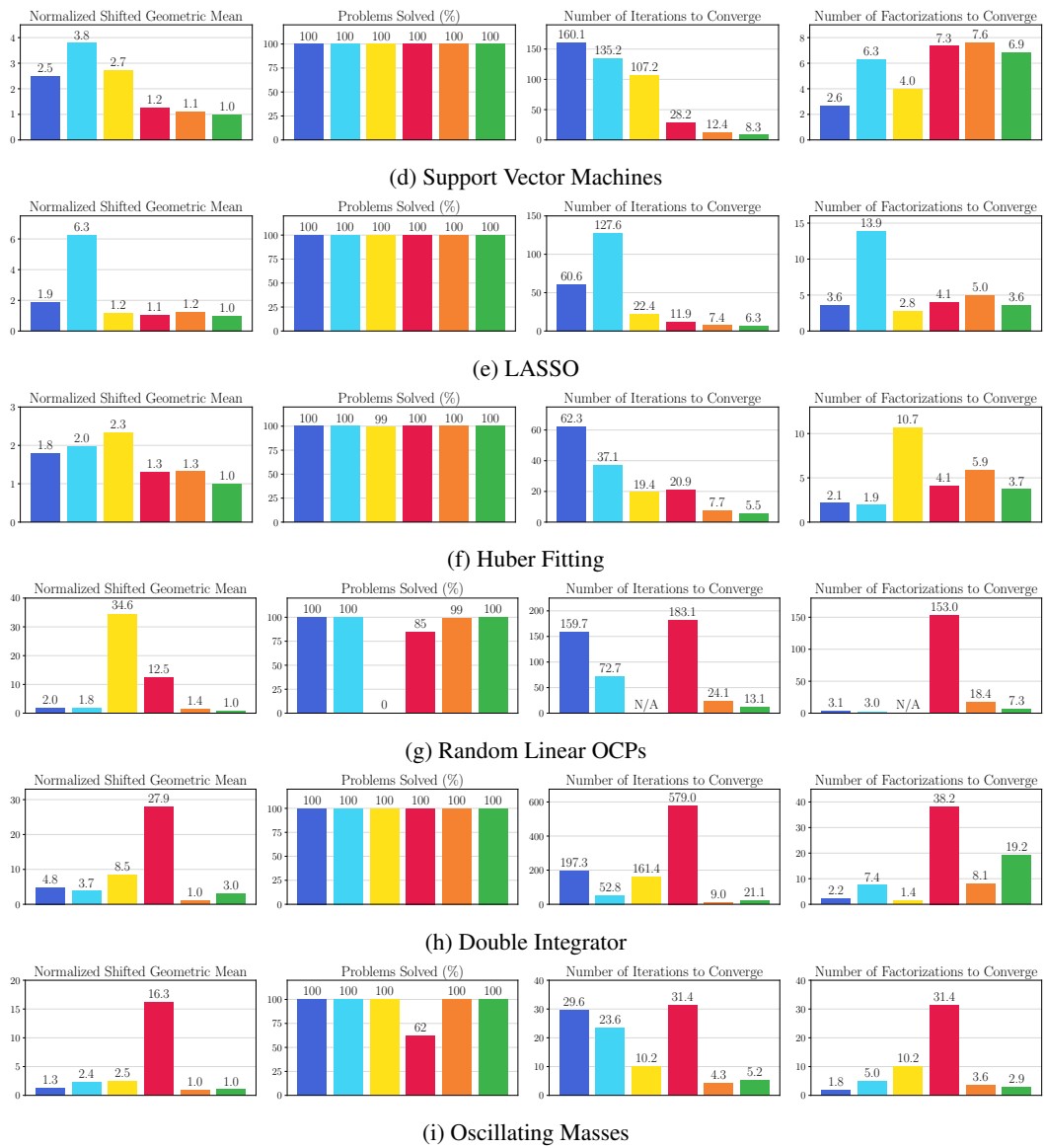

Figure 11: Performance comparison of vanilla vs. learned optimizers. Legend: OSQP, **FlexQP (Ours)**, Deep OSQP, Deep OSQP — RLQP Parameterization, Deep OSQP — Improved, **Deep FlexQP (Ours)**.

## K   PERFORMANCE COMPARISONS — LARGE-SCALE QPS

In this section, we report the full details of the large-scale QP experiments introduced in Section 5.2. Due to memory limitations, each optimizer is unfolded for 10 iterations and trained with a batch size of 1. Optimizers are trained for 5 epochs on 100 training problems and evaluated on 100 new test problems, with results presented in Fig. 12. Convergence is declared when the infinity norm of the QP residuals falls below an absolute tolerance of $\varepsilon = 10^{-3}$. Each optimizer is run until convergence or until a timeout of 10 minutes is reached. Similar to Appendix J, we report the normalized shifted geometric mean to better highlight the relative performance of each optimizer (wall clock times are provided in Fig. 6). We also report the percent of problems solved, the average number of iterations taken to converge, and the average number of CG iterations necessary to solve the linear system at each ADMM iteration. For the optimizers that failed to converge on any problems, we report the

average final QP residual infinity norms and average number of iterations run in Tables 2 and 3. This gives a rough idea of how far away the optimizers were from converging when they were timed out.

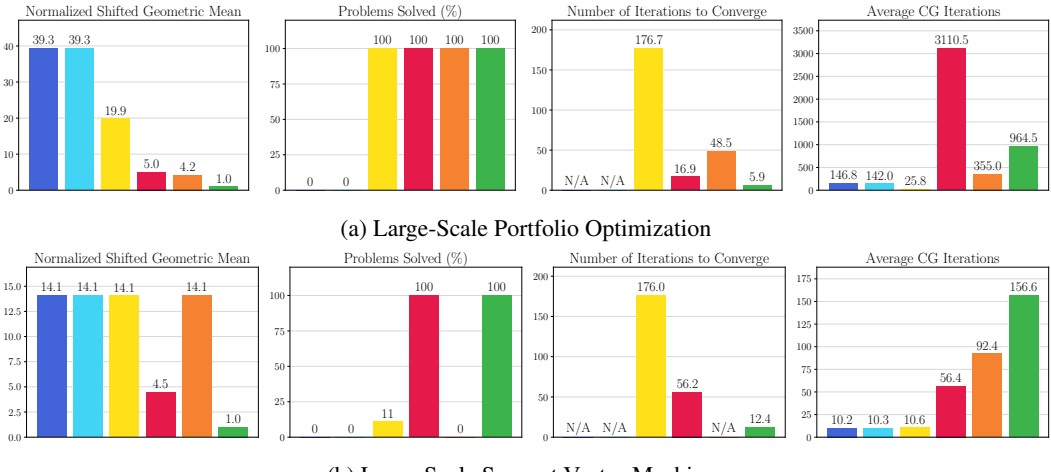

(a) Large-Scale Portfolio Optimization

(b) Large-Scale Support Vector Machines

Figure 12: Performance comparison of vanilla vs. learned optimizers. Legend: OSQP, FlexQP (Ours), Deep OSQP, Deep OSQP — RLQP Parameterization, Deep OSQP — Improved, Deep FlexQP (Ours).

| Optimizer | Final QP Residual Infinity Norm | Iterations Run |
|---|---|---|
| OSQP | 2.82e−3 | 325.5 |
| FlexQP (Ours) | 1.45e−3 | 326.3 |

Table 2: Large-scale portfolio optimization failed optimizer statistics.

| Optimizer | Final QP Residual Infinity Norm | Iterations Run |
|---|---|---|
| OSQP | 2.10e−3 | 178.9 |
| FlexQP (Ours) | 1.54e−3 | 326.3 |
| Deep OSQP — Improved | 8.04e−2 | 175.2 |

Table 3: Large-scale support vector machine failed optimizer statistics.

From these results, we observe that the traditional optimizers (OSQP and FlexQP), failed to converge on any of the problems within the allotted time. Furthermore, while the fine-tuning procedure worked for all learned optimizers on the portfolio optimization problems, it appears to have failed for Deep OSQP and Deep OSQP — Improved on the SVM problems.

Interestingly, the learned optimizers with the data-driven rules for the penalty parameters required many more CG iterations to solve their respective linear systems compared to the traditional optimizers. There seems to be a tradeoff between number of ADMM iterations required to converge vs. the condition number of the coefficient matrices generated at every ADMM iteration. For example, even though Deep FlexQP converges in relatively few iterations, each iteration requires an order of magnitude more CG iterations compared to OSQP to solve the linear systems to a similar precision. Using preconditioning with the learned optimizers would likely lead to even greater performance gains over the traditional optimizers.

## L  SAMPLE PARAMETER PREDICTION PLOTS

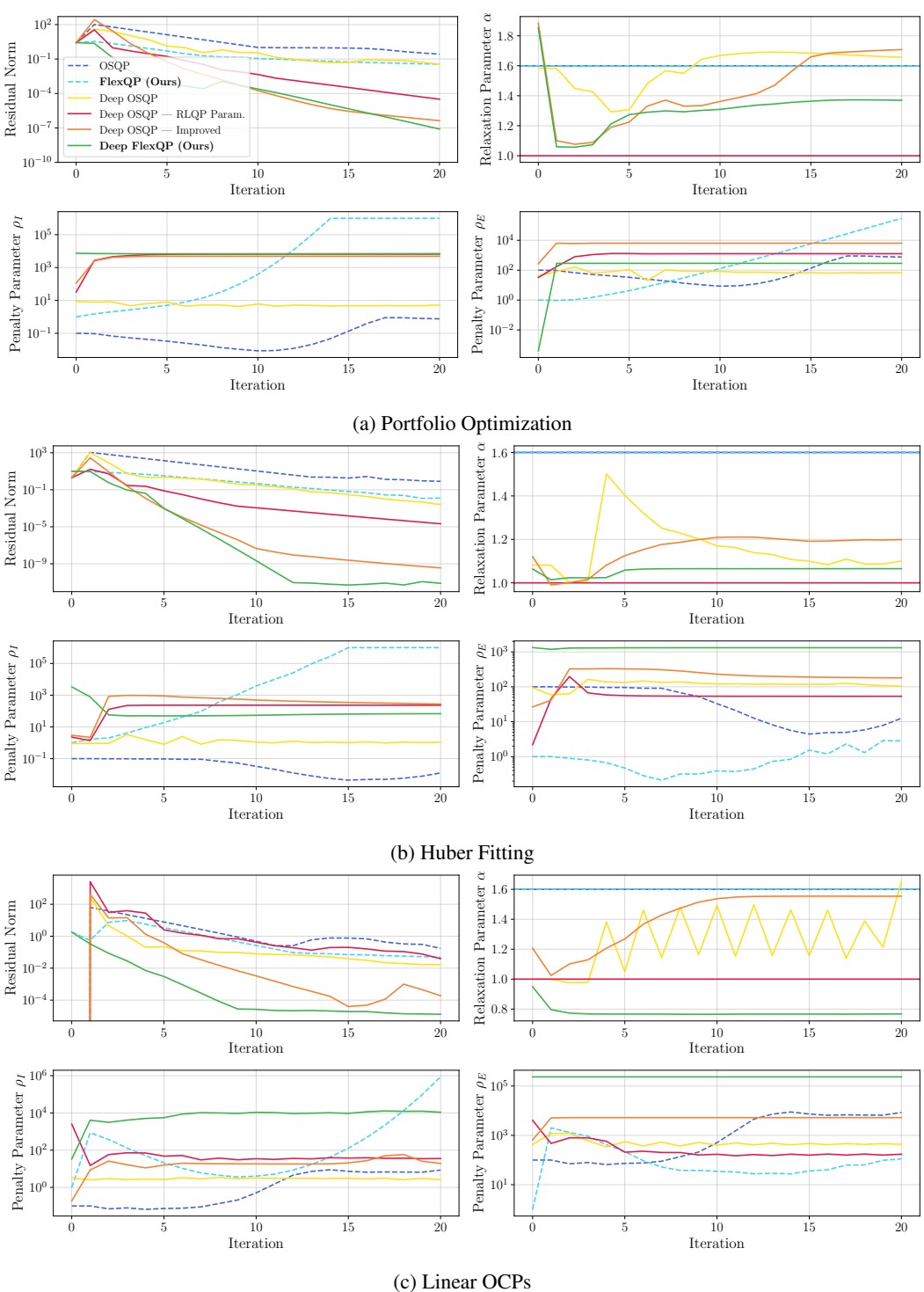

(a) Portfolio Optimization

(b) Huber Fitting

(c) Linear OCPs

Figure 13: Representative parameter predictions for a few problem instances.

In this section, we compare the $\alpha$, $\rho_I$, and $\rho_E$ predictions across the traditional and learned optimizers on a few representative optimization problems from Section 5.1. The parameters from the first 20 iterations of the different optimizers are shown in Fig. 13. Since Deep OSQP — RLQP

Parameterization, Deep OSQP — Improved, and Deep FlexQP output vectors of penalty parameters $\rho_I$ and $\rho_E$, we plot the mean prediction. Note that OSQP, FlexQP, and Deep OSQP — RLQP Parameterization use a fixed $\alpha$ of 1.6, 1.6, and 1.0, respectively. We also report the infinity norm of the QP residuals to get a better idea of when during the optimization a prediction is being made.

We observe that the learned rules seem to quickly adjust the parameters during the beginning of the optimization compared to the heuristic rules (OSQP and FlexQP), which seem to gradually adjust the parameters later in the optimization. Adjusting the parameters early might provide a beneficial effect on the convergence by allowing the optimizer to adjust to a better initial condition based on the first few evaluations of the problem residuals.

# M  SUPPLEMENTARY RESULTS FOR GENERALIZATION BOUNDS

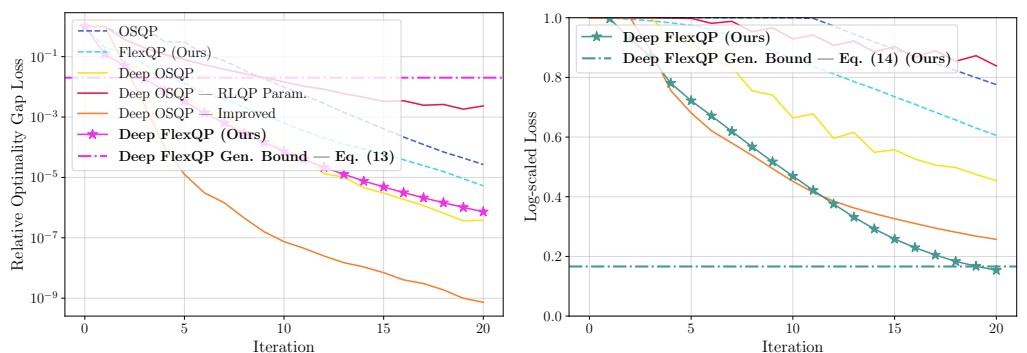

Figure 14: Generalization bounds for Deep FlexQP trained on 500 oscillating masses QPs. Following Fig. 4, this is another case where the generalization bound using the loss Eq. (13) is uninformative.

We provide an overview of the generalization bounds training procedure described by Saravanos et al. (2025), which in turn is adapted from the one described by Majumdar et al. (2021). Let $\boldsymbol{s} = (P, q, G, h, A, b, x^*, y_I^*, y_E^*) \sim \mathcal{D}$ denote a single sample from data distribution $\mathcal{D}$ and let $\mathcal{S} = \{\boldsymbol{s}_i\}_{i=1}^N$ be a dataset of $N$ samples. Let $\ell(\boldsymbol{s}, \theta) \in [0, 1]$ be a bounded loss for hypothesis $\theta \sim \mathcal{P}$. The true expected loss is defined as

$$\ell_{\mathcal{D}}(\mathcal{P}) = \mathbb{E}_{\boldsymbol{s} \sim \mathcal{D}} \mathbb{E}_{\theta \sim \mathcal{P}} [\ell(\boldsymbol{s}, \theta)], \tag{57}$$

and the sample loss is

$$\ell_{\mathcal{S}}(\mathcal{P}) = \mathbb{E}_{\theta \sim \mathcal{P}} \left[ \frac{1}{N} \sum_{i=1}^N \ell(\boldsymbol{s}_i, \theta) \right]. \tag{58}$$

We rely on the following PAC-Bayes bounds that hold with probability $1 - \delta$ (Majumdar et al., 2021, Corollary 1):

$$\ell_{\mathcal{D}}(\mathcal{P}) \leq \mathbb{D}^{-1} \left( \ell_{\mathcal{S}}(\mathcal{P}) || \frac{\mathbb{D}_{KL}(\mathcal{P} || \mathcal{P}_0) + \log(2\sqrt{N}/\delta)}{N} \right) \leq \ell_{\mathcal{S}}(\mathcal{P}) + \sqrt{\frac{\mathbb{D}_{KL}(\mathcal{P} || \mathcal{P}_0) + \log(2\sqrt{N}/\delta)}{2N}}, \tag{59}$$

where $\mathbb{D}^{-1}(p||c) = \sup\{q \in [0, 1] | \mathbb{D}_{KL}(\mathcal{B}(p) || \mathcal{B}(q)) \leq c\}$ is the inverse KL-divergence for Bernoulli random variables $\mathcal{B}(p)$ and $\mathcal{B}(q)$. The first bound is tighter and is therefore useful for computing the generalization bounds as a numerical certificate of performance, while the second bound has the nice interpretation of a training loss plus a regularization term that depends on the size of the training set and penalizes being off from the prior $\mathcal{P}_0$. During training, we minimize the second bound using either the loss Eq. (13) or Eq. (14).

During the evaluation of the tighter generalization bound (after training is complete), since it is difficult to compute the expectation over $\theta \sim \mathcal{P}$ in Eq. (58), we instead estimate the sample loss

using a large number of samples $\{\theta_j\}_{j=1}^M$ from $\mathcal{P}^*$:

$$\hat{\ell}_{\mathcal{S}}(\mathcal{P}^*) = \frac{1}{NM} \sum_{i=1}^N \sum_{j=1}^M \ell(\boldsymbol{s}_i, \theta_j). \tag{60}$$

The following sample convergence bound holds with probability $1 - \delta'$:

$$\bar{\ell}_{\mathcal{S}}(\mathcal{P}^*) \leq \mathbb{D}^{-1}\left(\hat{\ell}_{\mathcal{S}}(\mathcal{P}^*) || \frac{1}{M} \log(\frac{2}{\delta'})\right). \tag{61}$$

Combining these bounds results in a final version of the PAC-Bayes bound that holds with probability $1 - \delta - \delta'$ (Majumdar et al., 2021):

$$\ell_{\mathcal{D}}(\mathcal{P}^*) \leq \mathbb{D}^{-1}\left(\bar{\ell}_{\mathcal{S}}(\mathcal{P}^*) || \frac{\mathbb{D}_{KL}(\mathcal{P}^* || \mathcal{P}_0) + \log(2\sqrt{N}/\delta)}{N}\right). \tag{62}$$

This is the final bound that we report in our experiments.

The prior $\mathcal{P}_0$ for all our models is a stochastic Deep FlexQP trained for 500 epochs on 500 QPs generated from the **Random QPs with Equalities** problem class using the supervised learning setup from Section 4. We train Deep FlexQP for the generalization bound loss using either Eq. (13) or Eq. (14) with $\delta = 0.009$. We fix a training set for the generalization bounds using 500 problems from the class of interest and train the model for 1000 epochs. We evaluate Eq. (62) using $M = 10000$ model samples and $\delta' = 0.001$. Our bounds therefore hold with 99% probability. We report results comparing Eq. (13) vs. Eq. (14) for **LASSO** in Fig. 4 and for **Oscillating Masses** in Fig. 14. The test loss is computed over 1000 new samples from the target problem class. Overall, the loss Eq. (13) results in a less informative bound as it is above all the optimizers, even though the performance in practice can be much better.

Finally, we train Deep FlexQP on 500 QP subproblems generated from a nonlinear predictive safety filter task described in Appendix I using the same procedure to minimize the generalization bound through the loss defined in Eq. (14). We use this Deep FlexQP as the QP solver for the predictive safety filter SQP approach described in Appendix I.2. This provides a numerical certificate on the performance that would not hold for a traditional optimizer.

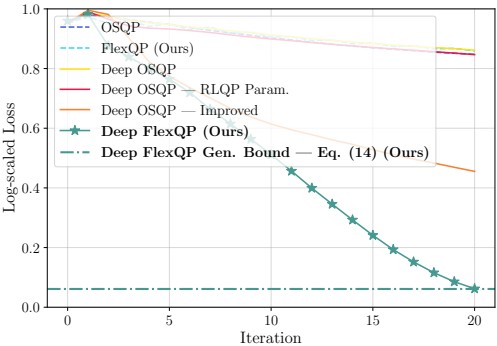

Figure 15: Generalization bound for Deep FlexQP trained on 500 QP subproblems generated from the nonlinear predictive safety filter task.

# N    LSTM vs. MLP Policy Comparison

This section presents an ablation analysis on the use of LSTMs to parameterize the policies in both Deep OSQP and Deep FlexQP. The use of LSTMs further leverages the analogy between deep-unfolded optimizers and RNNs, as discussed in Monga et al. (2021). Our hypothesis is that the RNN hidden state can encode a compressed history or context from the past optimization variables and residuals, thereby leading to a better prediction of the algorithm parameters to apply at the current iteration. Using an MLP only provides access to information from the current iterate, which could lead to a myopic choice of parameters.

Our results show that LSTMs enhance performance on several problem classes (Fig. 16). LSTMs appear to help the most on problems where the active constraints might switch suddenly from one iteration to the next. These types of problems include the machine learning ones, such as SVM, LASSO, and Huber fitting, as well as some of the control problems, like the oscillating masses.

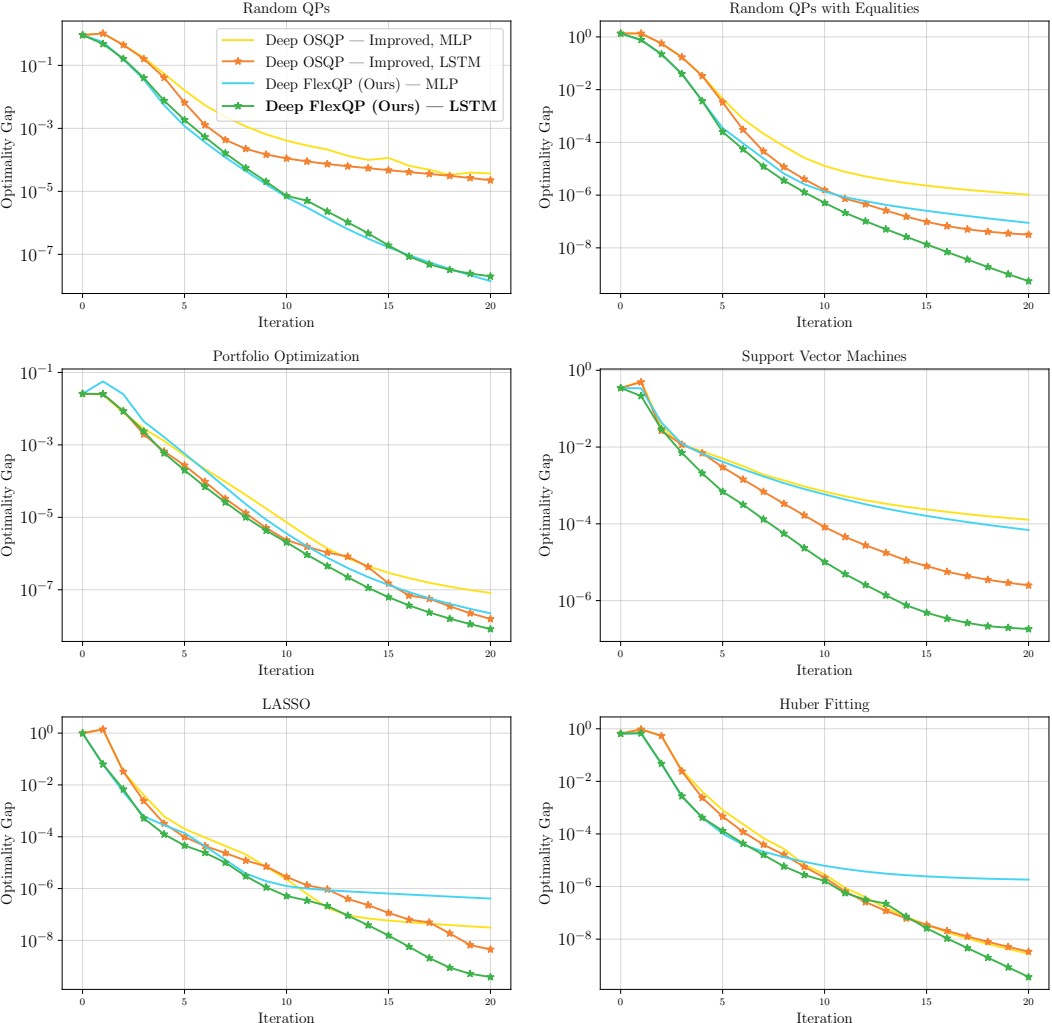

Figure 16: Comparison of MLP vs. LSTM policies. Performance is compared over 1000 test problems.

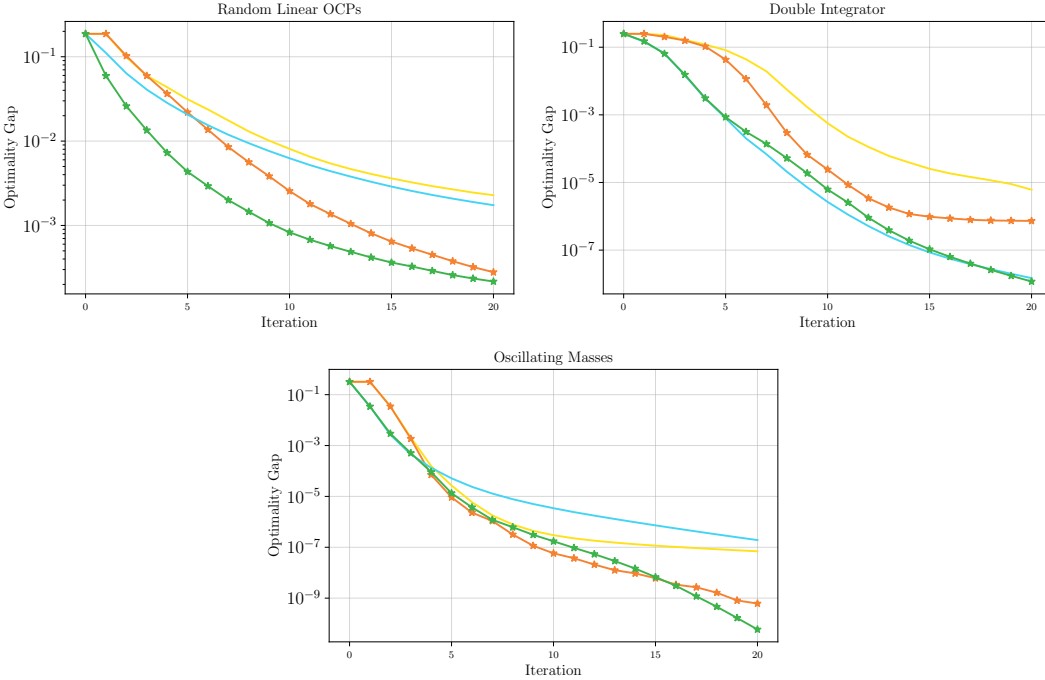

Figure 16: Comparison of MLP vs. LSTM policies. Performance is averaged over 1000 test problems.

## O  ABLATION ANALYSIS ON THE USE OF LAGRANGE MULTIPLIERS IN THE SUPERVISED LOSS

In Fig. 17, we compare the performance of Deep FlexQP on different problem classes using the optimality gap loss from Eq. (11) and our proposed loss Eq. (12). It is evident that our loss including the Lagrange multipliers outperforms the optimality gap loss in all cases, except for the oscillating masses problem class. This could be simply due to the fact that the performance is already approaching $10^{-10}$ at 20 iterations, and so small numerical differences play a bigger role. The overall increase in performance using the new loss can be explained by a stronger gradient signal given to Deep FlexQP to learn policies that ensure $\mu_I \geq \|y_I^*\|_\infty$ and $\mu_E \geq \|y_E^*\|_\infty$.

Surprisingly, however, the performance using both losses remains comparable. The ability of the optimality gap loss to perform nearly as well as the Lagrange multiplier loss likely stems from the coupling of $\mu_I$ with $\sigma_s, \rho_I$ and that of $\mu_E$ with $\rho_E$ in the Deep FlexQP architecture. That is, even with a weaker gradient signal from the optimality gap loss, the respective networks are able to learn a shared representation that allows effective learning of the penalty parameters $\mu$ as well.

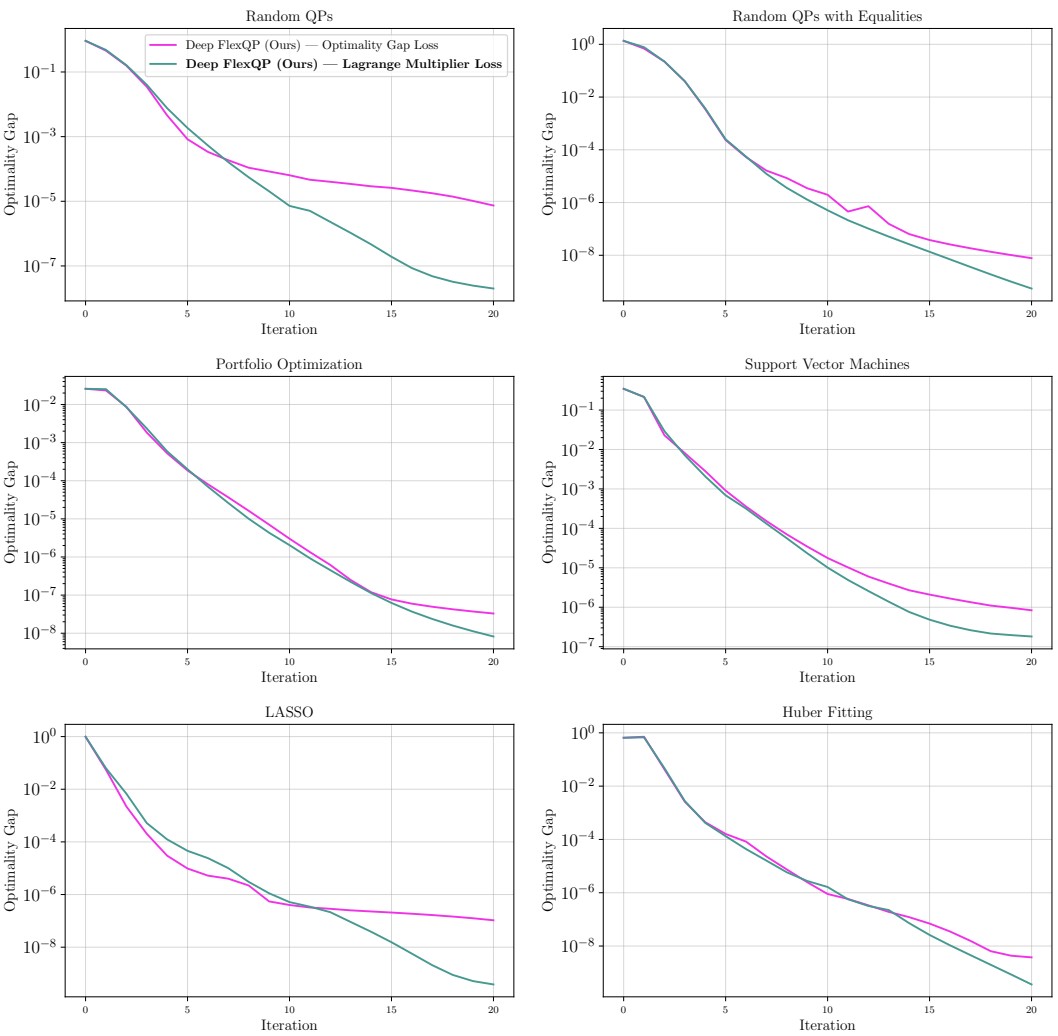

Figure 17: Comparison on using the optimality gap vs. Lagrange multiplier loss for training Deep FlexQP. Performance is averaged over 1000 test problems.

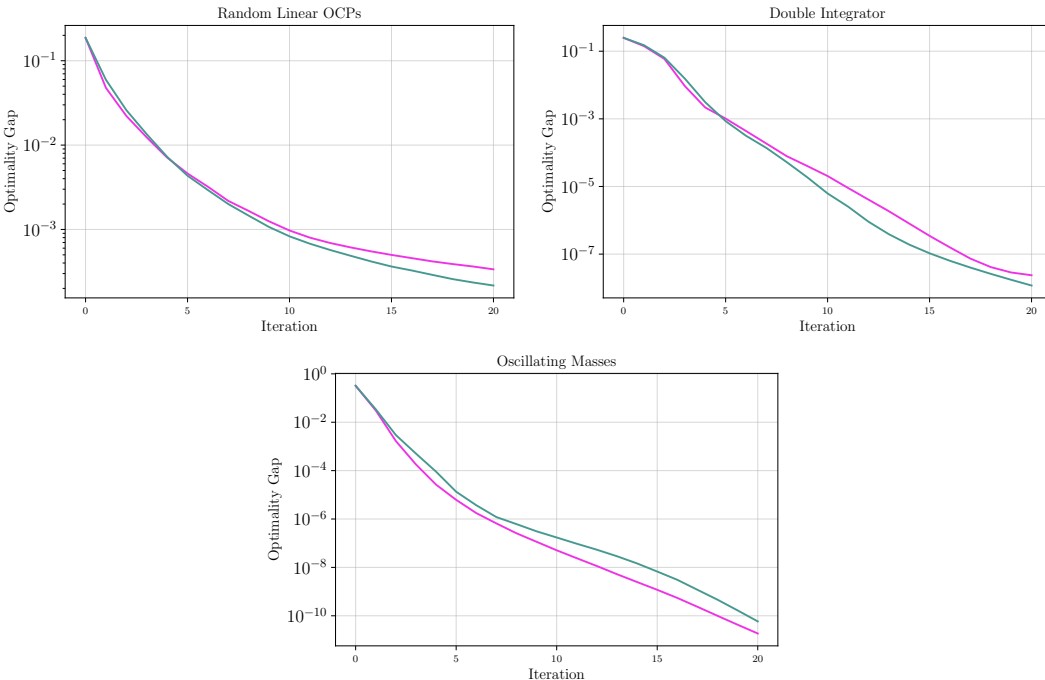

Figure 17: Comparison on using the optimality gap vs. Lagrange multiplier loss for training Deep FlexQP. Performance is averaged over 1000 test problems.

