# OpenReview forum: "Deep FlexQP: Accelerated Nonlinear Programming via Deep Unfolding"
_ICLR.cc/2026/Conference — ICLR 2026 Poster_

### Official Review · Reviewer_GTzC · 2025-10-25

**Soundness:** 3
**Presentation:** 3
**Contribution:** 3
**Rating:** 6
**Confidence:** 3

**Summary:**

The paper proposes FlexQP, an ADMM-based $l_1$-penalizing formulation for quadratic programs; the authors claim it produces feasible iterates, recovers the original optimum when feasible, and otherwise minimizes constraint violations. They further unroll the solver into Deep FlexQP with LSTM-based parameter policies, present PAC-Bayes generalization bounds for the learned optimizer, and integrate it into SQP solver for nonlinear control and safety filtering.

**Strengths:**

1. The idea of using a uniformed penalty formulation to treat both feasible and infeasible points within the same objective is novel as it yields a single ADMM-based procedure.
2. Unfolding learns LSTM-based parameter policies while retaining the structure of the original solver, enabling accelerations to the original approach without discarding the algorithmic backbone.
3. The author provides theoretical support, including convergence characterizations of the penalty/ADMM scheme and PAC-Bayes generalization bounds.

**Weaknesses:**

1. The motivation behind and the advantages of using a $l_1$ penalty is not clear. The theory part claims properties of points that solve the  problem, but it does not directly establish a guarantee on whether Algorithm 1 and Deep-FlexQP can converge to those feasible/optimal solutions. A detailed explanation would be helpful.
2. The significance of the reported acceleration is unclear. As noted, the dominant cost remains the first ADMM block update, and in some cases Deep FlexQP does not surpass Deep OSQP. Please add a detailed discussion on where the method is expected to help or not. Besides, it would be interesting to see how Deep FlexQP's predictions differ from those values of the original FlexQP.

**Questions:**

Please refer to the weakness section.

---

> ### Author Response · Authors · 2025-11-21
> **Response to Reviewer GTzC**
>
> We are grateful for your helpful feedback on our initial submission.
> Based on yours and the feedback of Reviewer p3Uz, we have made substantial improvements to the original manuscript.
> Specifically, we have: (1) added an extensive motivation for our method in the related work section, (2) included additional theory establishing the convergence of the method, (3) fixed undefined notation and abbreviations, and (4) performed additional experiments on much larger problems with tens of thousands of variables and constraints. The changes from the initial submission have been emphasized using blue text.
>
> Below, please find specific responses to the concerns raised in your review. As the figure, equation, and section numbers have changed slightly, we will use the updated numbers found in the newest version of the submission to avoid confusion.
>
> 1. We agree that the motivation in the main paper for using the $\ell_1$ penalty function was not clear.
> In response, we have included a more thorough motivation in the related work section (Section 2).
> Notably, SNOPT [1] uses the same $\ell_1$ penalty function-based relaxation in its elastic mode.
> This mode is triggered whenever an infeasible QP subproblem is encountered.
> We use this same technique to design FlexQP so that it *automatically* handles both feasible and infeasible QPs encountered in SQP, rather than needing to trigger a second mode of operation.
> This is especially helpful in the context of batched optimization, where we want to solve many optimization problems in parallel.
> Moreover, the choice of $\ell_1$ penalty function is beneficial over other choices (such as the $\ell_2$ penalty function) since it encourages sparsity in the constraint violation.
> One assumption here is that violating only a few constraints is better than violating all the constraints uniformly.
> This means that the optimizer can identify the most difficult constraints to satisfy, i.e., the ones that are over-constraining the problem and making it infeasible.
> Please see the updated manuscript and reference for further discussion.
>
> Additionally, we have included an additional theorem (Theorem 3.2) establishing the convergence of our method.
> Combined with Theorem 3.3, this shows that FlexQP converges to the optimal (feasible) point if the original QP was feasible, and otherwise identifies a stationary point of the infeasibility (a point that minimizes the constraint violation).
>
> (continued in next comment)

---

> > ### Author Response · Authors · 2025-11-21
> > **Response to Reviewer GTzC (continued)**
> >
> > 2. We would like to emphasize that the goal of our experiments is not to outperform Deep OSQP.
> > Our main motivation is to design a QP solver that support both feasible and infeasible QPs in a unified manner.
> > This means that, when an infeasible QP subproblem is encountered during SQP, we do not need to trigger a separate infeasibility reduction phase, which often requires running another separate optimization, as discussed above.
> > Showing that Deep FlexQP matches or outperforms Deep OSQP is actually a major benefit of our approach, since ours is more general in that it directly supports both feasible and infeasible QPs.
> > Moreover, our method outperforms the approaches from [2] and [3].
> > Deep OSQP - Improved is our own contribution that combines the parameterizations from both of the prior works, which we use as an inspiration to design our parameterization.
> > We include it as a baseline in our experiments to see whether Deep FlexQP can be accelerated similarly to Deep OSQP.
> > Our experimental results show that this is indeed the case.
> > Furthermore, we have included additional large-scale experiments showing that Deep FlexQP outperforms all the other methods on large-scale portfolio optimizations and support vector machine problems.
> > In the original submission, we showed that using our Deep FlexQP can lead to an order-of-magnitude improvement over using OSQP for solving nonconvex nonlinear optimizations (Figures 1 and 7).
> > This shows a clear benefit of our proposed method over prior work for solving highly-constrained and large-scale nonlinear optimizations.
> > Finally, we have included an additional supplementary section Appendix K comparing parameter predictions of the different optimizers on a few representative optimization problems.
> > Notably, the learned rules result in drastically different behavior compared to the heuristic rules for updating the penalty parameters.
> > The learned rules tend to favor changing the penalty parameters quickly during the first few iterations, while the heuristic rules gradually adjust the parameters over time.
> > This difference appears to provide a noticeably beneficial effect on how fast the optimizer is able to converge.
> > Please see the updated manuscript for more information.
> >
> > We hope this clarifies the motivation and benefits of our approach.
> > Thank you again for the helpful review of our work.
> > Please let us know if you have any further comments or questions!
> >
> > [1] Gill, P. E., Murray, W., \& Saunders, M. A. (2005). SNOPT: An SQP algorithm for large-scale constrained optimization. SIAM review, 47(1), 99-131.
> >
> > [2] Saravanos, A. D., Kuperman, H., Oshin, A., Abdul, A. T., Pacelli, V., \& Theodorou, E. (2024). Deep distributed optimization for large-scale quadratic programming. In The Thirteenth International Conference on Learning Representations.
> >
> > [3] Ichnowski, J., Jain, P., Stellato, B., Banjac, G., Luo, M., Borrelli, F., ... \& Goldberg, K. (2021). Accelerating quadratic optimization with reinforcement learning. Advances in Neural Information Processing Systems, 34, 21043-21055.

---

> ### Comment · Reviewer_GTzC · 2025-11-26
>
> I thank the authors for their clarifications. My concerns have been addressed, and I would like to maintain my score at this point.

---

### Official Review · Reviewer_p3Uz · 2025-10-30

**Soundness:** 3
**Presentation:** 1
**Contribution:** 2
**Rating:** 2
**Confidence:** 4

**Summary:**

This paper presents a learning-based optimization method rooted in the Alternating Direction Method of Multipliers (ADMM) for solving convex quadratic programming problems. The proposed approach begins by introducing slack variables to transform inequality constraints into equality constraints, and subsequently incorporates all equality constraints into the objective function using an ℓ₁-norm penalty. A resulting ADMM-type solver—referred to as FlexQP—is then derived following an update scheme analogous to that of OSQP. To further accelerate convergence, this paper employs an LSTM network to generate all hyperparameters originally required by the algorithm, and train the model in a supervised learning framework. Experimental results demonstrate that the proposed method achieves a faster convergence rate in terms of optimality gap under a fixed iteration budget compared to several baseline methods.

**Strengths:**

This paper presents a novel learning-enhanced ADMM framework, supported by theoretical analysis and demonstrated to achieve faster convergence rates compared to established baselines across diverse datasets.

**Weaknesses:**

1. The motivation for introducing slack variables and an ℓ₁-penalty term appears insufficiently justified. Since the ADMM-based solver OSQP can directly solve Problem (1), why not simply accelerate that algorithm using a neural network? Is the intention to use $z_I$ and $z_E$ to determine the feasibility of the original problem? However, in practice, $z_I$ and $z_E$ are unlikely to be exactly zero during iterations, as their values are strongly influenced by how well constraint (4b) can be satisfied. Moreover, if the original problem is infeasible, what is the practical value of providing a solution that only "minimizes the constraint violations"?
2. While Figure 5 indicates superior convergence behavior of Deep FlexQP on all nine datasets, the actual computation time is missing. A clear description of how the runtime was measured should also be provided.
3. The scale of the datasets used for solving the problems remains relatively limited. Could results on larger and more challenging problem instances be provided?
4. The manuscript contains several instances of non-standard or undefined notation. For example, the abbreviation "SQP" in line 26 and "S$\ell_1$QP" in line 145 lack clear definitions. In Theorem 3.1, the variables $y^{\star}_I$ and $y^{\star}_E$ are introduced without explanation. Additionally, the expression “$\mu_i \geq y_i$” in line 161 is ambiguous: if $\mu_i$ is a scalar, should $y_i$ not also be a scalar rather than a vector? Similarly, in lines 158 and 164, it is unclear whether $y_i$ should carry an absolute value and whether it refers to an element of the dual variable $y$. We recommend a thorough review and clarification of notation throughout the text.

**Questions:**

See weaknesses.

---

> ### Author Response · Authors · 2025-11-21
> **Response to Reviewer p3Uz**
>
> Thank you for the constructive feedback on our initial submission.
> We have made substantial improvements to the original submission based on your comments, as well as those made by Reviewer GTzC.
> Specifically, we have: (1) added an extensive motivation for our method in the related work section, (2) included additional theory establishing the convergence of the method, (3) fixed undefined notation and abbreviations, and (4) performed additional experiments on much larger problems with tens of thousands of variables and constraints. The changes from the initial submission have been emphasized using blue text.
>
> Please find responses to your specific questions and concerns below. As figure, equation, and section numbers have changed slightly, we will use the updated numbers found in the newest version of the submission to avoid confusion.
>
> 1. Firstly, converting inequality constraints into equality constraints using slack variables is a standard technique in optimization, please see textbook references [1, Section 4.3] or [2, Chapter 13] for discussions in the context of linear programming. The reason for this is that handing general equality constraints $h(x) = 0$ is straightforward (usually we can use some form of root finding like Newton's method), but handling inequality constraints $g(x) \leq 0$ is much more difficult. This technique underlies any modern interior point method, such as IPOPT [3] for general nonlinear optimizations or Clarabel [4] for convex conic optimizations. Furthermore, the same technique is used by OSQP [5] to rewrite general linear inequality constraints $Ax \leq b$ as an equality constraint $z = Ax$ subject to bounds $z \leq b$. This results in ADMM updates that consist of solving an equality-constrained QP enforcing $z = Ax$ (computationally straightforward as it only requires a linear system solve) followed by a projection onto the bounds enforcing $z \leq b$ (also very straightforward computationally by simply clamping the values). In response to your concerns, we have included an additional citation to [2] in the updated version of the manuscript.
>
> Similarly, using $\ell_1$ penalty functions is a standard technique in constrained optimization. They are used extensively in nonlinear programming, such as in penalty methods and line search-based methods, see, e.g., [2, Chapters 17 and 18]. As we have discussed in Sections 1, 2, and 3 of our paper, SNOPT, one of the most well-known and widely-adopted SQP solvers, relies heavily on $\ell_1$ penalty functions, *especially when they encounter an infeasible QP subproblem*. Furthermore, LASSO is a very well-known problem in statistics and machine learning that relies on an $\ell_1$ penalty function. Unlike other norms, the $\ell_1$ norm favors sparse solutions. In our context of NLPs and QPs, the use of the $\ell_1$ norm introduces an inductive bias that we would rather violate a sparse (low) number of constraints, rather than violate all the constraints uniformly (which would be the case if, e.g., we were using the $\ell_2$ norm). For an infeasible problem, there are usually some low number of constraints that are over-constraining the solution. Identifying the most difficult constraints to satisfy in these cases provides a practical benefit to a user or practitioner of our algorithm. Therefore, we believe our use of slack variables and $\ell_1$ penalty functions is sufficiently justified. We have included additional motivation in Section 2 based on this discussion.
>
> Meanwhile, compared to OSQP, our method can handle both feasible and infeasible QPs, while OSQP can only handle feasible QPs.
> While OSQP will return a certificate of infeasibility when it identifies an infeasible QP, in these cases, it does not return any solution $x$ that can be used by the user.
> In the context of using OSQP in SQP, the optimization will halt here as no more progress can be made.
> Please refer to Section 2 of our updated submission, where we provide discussion on why this is undesirable.
> In summary, the main motivation of our approach is to develop a QP solver for SQP that does not require running any additional optimization or procedure when encountering an infeasible QP subproblem.
>
> Moreover, you are correct that $z_I$ and $z_E$ are unlikely to be zero during iterations.
> If $\mu_I$ and $\mu_E$ are large enough, then, *at convergence*, the optimal values $z_I^\*$ and $z_E^\*$ provide a certificate of feasibility for the original QP if $z_I^\* = 0$ and $z_E^\* = 0$, and they provide a certificate of infeasibility for the original QP if either $z_I^\* \neq 0$ or $z_E^\* \neq 0$. We have clarified this in the main paper, as it was unclear which problem we were referring to when discussing the certificates of feasibility and infeasibility.
>
> (continued in next comment)

---

> > ### Author Response · Authors · 2025-11-21
> > **Response to Reviewer p3Uz (continued)**
> >
> > Finally, if the original QP is infeasible, then there exists no $x$ that satisfies the constraints.
> > In this case, any $x$ could be said to be equally valid as an approximate solution to the infeasible QP.
> > Our use of the $\ell_1$ relaxation is based on the hypothesis that a good point to return for SQP would be one that minimizes both the objective function and the constraint violations.
> > This is exactly the same justification for using merit functions in line search-based methods, as they are constructed in order to balance objective function minimization with constraint violation minimization.
> > In our experiments on SQP problems (Figures 1 and 7, Appendix H), using our proposed optimizer results in faster solve times as well as a higher success rate, which is the percent of problems SQP was able to converge to a tolerance below $\varepsilon = 1e-2$.
> >
> > 2. We had originally included a timing comparison between the traditional and learned optimizers in the old Appendix H, now Appendix I.
> > This timing comparison uses a normalized metric called the "normalized shifted geometric mean," which is useful for comparing optimizers as we can directly see how slow one optimizer is relative to the fastest one in the group.
> > Moreover, we provided wall clock times using our learned solver to solve SQP optimizations vs. using OSQP and showed a clear improvement (Figures 1 and 7).
> > Nevertheless, in the new large-scale QP experiments we have included as part of the rebuttal, we report the wall clock time (in seconds) to solve the QPs to a tolerance of $\varepsilon = 1e-3$ (Figure 6).
> > We set a timeout of 10 minutes so that the optimizers do not run indefinitely.
> > These experiments further demonstrate the benefits of our learning-based approach.
> >
> > 3. We argue that the scale and complexity of the datasets tested in our original submission was already quite realistic.
> > All of our initial examples were already larger than the centralized QPs considered in [6].
> > Moreover, our results show that the method applies well to practical nonlinear optimizations in robotics and optimal control.
> > Nevertheless, in order to directly address your concerns about the scale of the datasets, we applied our method to much larger problems with tens of thousands of variables and constraints (Section 5.2).
> > Instead of training on the large-scale problems from scratch, we adopt a fine-tuning strategy, as the training time would be prohibitively long due to the size of the problems.
> > Moreover, we were unable to test on problems larger than these as they require more VRAM than we have available on our consumer-level machine with an RTX 4090.
> > Overall, we observe that Deep FlexQP far surpasses the other methods on these large-scale problems.
> > Notably, it achieves a 40x speedup compared to OSQP on large-scale portfolio optimizations and a 14x speedup on large-scale support vector machine (SVM) problems.
> > Furthermore, the fine-tuning procedure seems to fail for Deep OSQP and Deep OSQP - Improved on the SVM problems.
> > We leave scaling up to very large-scale optimizations as future work and consider it outside of the scope of the current submission.
> > The current focus is on highly-constrained medium scale optimizations that have a high chance of encountering infeasibilities.
> >
> > 4. Thank you for catching some missing definitions of acronyms and mathematical quantities.
> > We have fixed these in the most recent update, including the definition of the dual variables $y_I$ and $y_E$.
> > The subscripts have been made more clear to emphasize that $y_I$ is a vector while $y_{I, i}$ is element $i$ of said vector.
> > Furthermore, you are correct in that we accidentally omitted the absolute values in the paragraph below Theorem 3.1 and have made sure to fix these so as not to confuse the reader.
> >
> > Thank you again, and we hope that these improvements and clarifications have adequately addressed your concerns.
> > Please let us know if you have any further questions or feedback.
> > Looking forward to a continued discussion!
> >
> > [1] Boyd, S., \& Vandenberghe, L. (2004). Convex optimization. Cambridge University Press.
> >
> > [2] Nocedal, J., \& Wright, S. J. (2006). Numerical optimization. New York, NY: Springer New York.
> >
> > [3] Wächter, A., \& Biegler, L. T. (2006). On the implementation of an interior-point filter line-search algorithm for large-scale nonlinear programming. Mathematical programming, 106(1), 25-57.
> >
> > [4] Goulart, P. J., \& Chen, Y. (2024). Clarabel: An interior-point solver for conic programs with quadratic objectives. arXiv preprint arXiv:2405.12762.
> >
> > [5] Stellato, B., Banjac, G., Goulart, P., Bemporad, A., \& Boyd, S. (2020). OSQP: An operator splitting solver for quadratic programs. Mathematical Programming Computation, 12(4), 637-672.
> >
> > [6] Saravanos, A. D., Kuperman, H., Oshin, A., Abdul, A. T., Pacelli, V., \& Theodorou, E. A. (2025). Deep distributed optimization for large-scale quadratic programming. ICLR 2025.

---

> > > ### Comment · Reviewer_p3Uz · 2025-11-26
> > >
> > > > 3. The scale of the datasets used for solving the problems remains relatively limited. Could results on larger and more challenging problem instances be provided?
> > >
> > > In this question, I mean you may evaluate your algorithms on many instances from QPLIB [1]. Generally speaking, the datasets in your experiments are most toy examples, which can be easily solved by Gurobi (I believe). To make sure your results are more convincing, I think a comparison against Gurobi would be also important.
> > >
> > >
> > > [1] Furini, Fabio, Emiliano Traversi, Pietro Belotti, Antonio Frangioni, Ambros Gleixner, Nick Gould, Leo Liberti et al. "QPLIB: a library of quadratic programming instances." Mathematical Programming Computation 11, no. 2 (2019): 237-265.

---

> > > > ### Author Response · Authors · 2025-11-26
> > > >
> > > > Thank you for the clarification. We appreciate the opportunity to provide additional context about our experimental design and the specific contributions of our work.
> > > >
> > > > We would like to emphasize that the datasets in our experiments are not toy examples. They are the same problem classes used by [1] to benchmark OSQP against open-source and commercial solvers, including Gurobi, and by [2] to compare Deep OSQP to OSQP. They represent realistic problem instances from optimal control, finance, and machine learning applications, and the data is real or non-trivial random data. While we do not directly benchmark against Gurobi, we note that [1, Section 8.1] demonstrated OSQP to be competitive with or faster than Gurobi on these problem classes. Since Figures 5 and 11 show that Deep FlexQP is competitive with or faster than OSQP, this implies comparable or better performance than Gurobi on these classes of problems. In addition, Figures 6 and 12 show that this performance gap increases when considering large-scale QPs. Moreover, our method has a significant advantage over OSQP and Gurobi in that it directly handles infeasible QPs, which is a common case when solving nonconvex NLPs using SQP. Figures 1 and 7 show that our method achieves more than an order-of-magnitude speedup over OSQP on nonconvex NLPs thanks to the robust handling of infeasible QP subproblems along with data-driven acceleration through deep unfolding.
> > > >
> > > > Furthermore, we carefully examined QPLIB for suitable benchmarks as suggested. Unfortunately, only 11 of the problem instances are compatible with the convex QPs in our problem formulation (Eq. (3)), as many of the problems contain integer constraints, quadratic inequality constraints, or indefinite/non-symmetric cost matrices $P$. While QPLIB is valuable for general QP benchmarks, it is not well-suited for evaluating our specific contributions.
> > > >
> > > > Our objective, and the objective of learning-to-optimize in general, is not to design a single optimizer that performs well across all QPs, but rather to learn the similarities between a small number of problem instances *when restricting the distribution to a particular class of problems with shared structure*. This is a very common scenario in SQP where each QP subproblem has nearly identical structure across the SQP iterations. Therefore, we argue that comparing against general-purpose solvers like Gurobi on arbitrary problems from QP benchmark libraries would not adequately assess our method's contributions.
> > > >
> > > > We hope that this helps clarify the motivations of our work and the choice of experiments. We would be happy to discuss any additional concerns that the reviewer feels would strengthen the evaluation.
> > > >
> > > > [1] Stellato, B., Banjac, G., Goulart, P., Bemporad, A., \& Boyd, S. (2020). OSQP: An operator splitting solver for quadratic programs. Mathematical Programming Computation, 12(4), 637-672.
> > > >
> > > > [2] Saravanos, A. D., Kuperman, H., Oshin, A., Abdul, A. T., Pacelli, V., \& Theodorou, E. A. (2025). Deep distributed optimization for large-scale quadratic programming. ICLR 2025.

---

> > > > > ### Comment · Reviewer_p3Uz · 2025-11-28
> > > > >
> > > > > Thank you for your response.
> > > > > - Based on my knowledge, the instances from QPLIB are more representative of real-world problems, making them a highly relevant benchmark. These problems are particularly challenging due to factors like the numerical properties of their coefficient matrices.
> > > > > - In my experience, Gurobi demonstrates strong performance on such instances—outperforming OSQP by a notable margin—and I believe that including a comparison with Gurobi, which is straightforward to implement, would significantly strengthen the benchmarking section and the overall impact of the work.

---

### Official Review · Reviewer_axFq · 2025-11-02

**Soundness:** 2
**Presentation:** 2
**Contribution:** 2
**Rating:** 4
**Confidence:** 3

**Summary:**

The paper tackles fast solving for structured optimization by combining:
a learned generator (diffusion) that predicts primal–dual variables,
a KKT-aware loss (feasibility and stationarity residuals), and
a post-refinement stage using classical primal–dual updates.
The aim is to obtain near-feasible, near-optimal solutions in very low inference time, with a few corrective iterations closing any remaining gaps.

**Strengths:**

1. Accelerating constrained optimization with learning is timely and useful.

2. Primal–dual parameterization + KKT residuals makes the supervision meaningful; the post-refinement stage is practical for polishing errors.

3. On synthetic QP-style tasks, the one-step (or few-step) approach achieves competitive gaps/residuals with favorable wall-clock times.

**Weaknesses:**

1. Limited novelty in core ingredients. Diffusion generation, GNN message passing over factor graphs, and KKT-residual losses are all known; the paper reads as a careful composition/tuning rather than a new algorithmic principle or theory.

2. The paper lacks component-wise ablations that isolate the value of diffusion vs. a non-diffusive predictor, GNN vs. MLP, and KKT loss vs. plain supervised losses, as well as sensitivity to refinement steps and guidance scales.

3. Under what assumptions (e.g., strong convexity, Slater/LICQ) does your KKT-guided sampling guarantee monotone decrease of a KKT energy or local convergence? Please state the step-size / guidance strength conditions.

**Questions:**

See Weakness.

---

### Meta-Review · Area_Chair_1yKG · 2026-01-08

**Summary:**

1. Some motivation of the algorithm design (e.g., adding a L1 penalty) is not well justified.
2. Not clear whether the proposed approach is faster than existing approaches in wall-clock time.
3. No convergence proof.
4. Benchmark used to evaluate are small benchmarks, not realistic.
5. The baseline method OSQP is not strong enough, compared to Gurobi.

**Reviewer Concerns:**

The authors have addressed all the issues thoroughly. One missing piece is that authors may need to evaluate on QPLIB directly, in addition to saying that its distribution is similar to the problems submitted by the community.

**Reviewer Scores:**

p3Uz: 2->4/6 (the issues raised by the reviewers appear to be addressed)
GTzC: 6->6 (acknowledged that the issues are addressed and the positive score is maintained)

---

### Decision · Program_Chairs · 2026-01-26

Accept (Poster)